# Multi-Armed Bandits with Network Interference

**Abhineet Agarwal**
Department of Statistics
UC Berkeley
aa3797@berkeley.edu

**Anish Agarwal**
Department of IEOR
Columbia University
aa5194@columbia.edu

**Lorenzo Masoero**
Amazon
masoerl@amazon.com*

**Justin Whitehouse**
Computer Science Department
Carnegie Mellon University
jwhiteho@andrew.cmu.edu

## Abstract

Online experimentation with interference is a common challenge in modern applications such as e-commerce and adaptive clinical trials in medicine. For example, in online marketplaces, the revenue of a good depends on discounts applied to competing goods. Statistical inference with interference is widely studied in the offline setting, but far less is known about how to adaptively assign treatments to minimize regret. We address this gap by studying a multi-armed bandit (MAB) problem where a learner (e-commerce platform) sequentially assigns one of possible $\mathcal{A}$ actions (discounts) to $N$ units (goods) over $T$ rounds to minimize regret (maximize revenue). Unlike traditional MAB problems, the reward of each unit depends on the treatments assigned to other units, i.e., there is *interference* across the underlying network of units. With $\mathcal{A}$ actions and $N$ units, minimizing regret is combinatorially difficult since the action space grows as $\mathcal{A}^N$. To overcome this issue, we study a *sparse network interference* model, where the reward of a unit is only affected by the treatments assigned to $s$ neighboring units. We use tools from discrete Fourier analysis to develop a sparse linear representation of the unit-specific reward $r_n : [\mathcal{A}]^N \to \mathbb{R}$, and propose simple, linear regression-based algorithms to minimize regret. Importantly, our algorithms achieve provably low regret both when the learner observes the interference neighborhood for all units and when it is unknown. This significantly generalizes other works on this topic which impose strict conditions on the strength of interference on a *known* network, and also compare regret to a markedly weaker optimal action. Empirically, we corroborate our theoretical findings via numerical simulations.

## 1 Introduction

Online experimentation is an indispensable tool for modern decision-makers in settings ranging from e-commerce marketplaces [Li et al., 2016] to adaptive clinical trials in medicine [Durand et al., 2018]. Despite the wide-spread use of online experimentation to assign treatments to units (e.g., individuals, subgroups, or goods), a significant challenge in these settings is that outcomes of one unit are often affected by treatments assigned to other units. That is, there is *interference* across the underlying network of units. For example, in e-commerce, the revenue for a given good depends on discounts applied to related or competing goods. In medicine, an individual's risk of disease depends not only on their own vaccination status but also on that of others in their network.

---

*The research presented in this paper was conducted independently and is entirely unrelated to the author's current appointment at Amazon.

38th Conference on Neural Information Processing Systems (NeurIPS 2024).

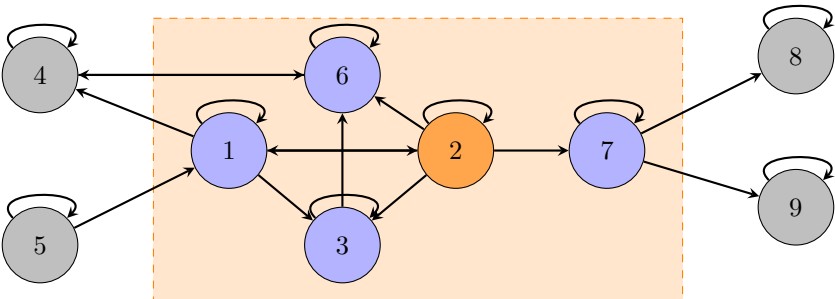

**Figure 1:** A visual representation of sparse network interference. In this toy example, we have $N = 9$ units, and visualize the interference pattern. For unit 2 (orange), its outcomes are affected by the treatments of its neighbours (blue) $\mathcal{N}(2) = \{1, 2, 3, 6, 7\}$.

Network interference often invalidates standard tools and algorithms for the design and analysis of experiments. While there has been significant work done to develop tools for statistical inference in the offline setting (see Section 2), this problem has mostly been unaddressed in the online learning setting. In this paper, we address this gap by studying the multi-armed bandit (MAB) problem with network interference. We consider the setting where a learner (online marketplace) assigns one of possible $\mathcal{A} \in \mathbb{N}$ actions (varying discounts) to $N$ units (goods) over $T$ rounds to minimize average regret. In our setting, the reward of a unit $n \in [N] := \{1, \ldots, N\}$ depends on the actions assigned to other units.[2] With $N$ units and $\mathcal{A}$ actions, achieving low regret is difficult since there are $\mathcal{A}^N$ possible treatment assignments. Naively applying typical MAB methods such as the upper confidence bound (UCB) algorithm [Auer et al., 2002] leads to regret that scales as $O(\sqrt{\mathcal{A}^N T})$, which can be prohibitively large due to the exponential dependence on $N$. Further, without any assumptions on the interference pattern, regret scaling as $\widetilde{\Omega}(\sqrt{\mathcal{A}^N T})$ is unavoidable due to lower bounds from the MAB literature [Lattimore and Szepesvári, 2020].

To overcome this issue, we consider a natural and widely-studied model of *sparse network interference*, where the reward $r_n : [\mathcal{A}]^N \to \mathbb{R}$ for unit $n$ is affected by the treatment assignment of at most $s$ other units, i.e., neighbours. See Figure 1 for a visualization. Under this model, we provide algorithms that provably achieve low regret both when the learner observes the network (i.e., the learner knows the $s$ neighbors for all units $n$), and when it is unknown. Our results allow for more general interference patterns and define regret with respect to a significantly stronger comparator policy than existing results in the literature.

**Contributions.**

(i) For each unit $n \in [N]$, we use the Fourier analysis of discrete functions to re-express its reward $r_n : [\mathcal{A}]^N : \to \mathbb{R}$ as a linear function in the Fourier basis with coefficients $\boldsymbol{\theta}_n \in \mathbb{R}^{\mathcal{A}^N}$. We show sparse network interference implies $\boldsymbol{\theta}_n$ is $\mathcal{A}^s$ sparse for all $n \in [N]$. This sparse linear representation motivates a simple 'explore-then-commit' style algorithm that uniformly explores actions, then fits a linear model to estimate unit-specific rewards (i.e., $\boldsymbol{\theta}_n$).

(ii) With known interference (i.e., the learner knows the $s$ neighbors for all $n \in [N]$), our algorithm exploits this knowledge to estimate $r_n$ by performing ordinary least squares (OLS) locally (i.e., per unit) on the Fourier basis elements where $\boldsymbol{\theta}_n$ is non-zero. Our analysis establishes regret $\tilde{O}((\mathcal{A}^s T)^{2/3})$ for this algorithm.

(iii) With unknown interference, we use the Lasso instead of OLS locally which adapts to sparsity of $\boldsymbol{\theta}_n$ and establish regret $\tilde{O}(N^{1/3} (\mathcal{A}^s T)^{2/3})$. We argue this $T^{2/3}$ scaling cannot be improved.

(iv) Numerical simulations with network interference show our method outperforms baselines.

## 2 Related Work

**Causal inference and bandits with interference.** The problem of learning causal effects in the presence of cross-unit interference has received significant study from the causal inference com-

---

[2]For any positive integer $x$, we let $[x] := \{1, \ldots x\}$.

munity (see [Bajari et al., 2023] for a thorough overview). Cross-unit interference violates basic assumptions for causal identifiability, invalidating standard designs and analyses.[3] As a result, authors have developed methodologies for estimating causal effects under several models of interference such as intra-group interference [Hudgens and Halloran, 2008, Rosenbaum, 2007], interference neighborhoods [Gao and Ding, 2023, Ugander et al., 2013, Bhattacharya et al., 2020, Yu et al., 2022, Cen et al., 2022], in bipartite graphs representative of modern online markets [Pouget-Abadie et al., 2019, Bajari et al., 2021, 2023], in panel data settings [Agarwal et al., 2022] as well as under a general model of interference, generally encoded via "exposure mappings" [Aronow, 2012, Aronow and Samii, 2017]. Despite this large literature, there is much less work on learning with interference in online settings. Jia et al. [2024] take an important step towards addressing this gap by studying MABs with network interference, but assume a known, grid-like interference pattern, where the strength of the interference decays as the $\ell_1$ distance between units grows. Moreover, their focus – unlike ours – is on establishing regret rates with respect to the best constant policy, i.e. the best policy that assigns each unit the same treatment. We also note that the authors consider a setting more closely aligned with the adversarial bandit literature, whereas the results in this paper are closer to those in the stochastic bandit literature. See Section 3 for a detailed description of these differences.

**Bandits with high-dimensional action spaces.** In MAB problems, regret is typically lower bounded by $\widetilde{\Omega}(\sqrt{\#\text{Actions} \cdot T})$, where $\#\text{Actions} = \mathcal{A}^N$ in our setting. Typically, this curse of dimensionality is addressed by sparsity constraints on the rewards, where only a small fraction of actions have non-zero rewards [Kwon et al., 2017, Abbasi-Yadkori et al., 2012, Hao et al., 2020]. Particularly relevant to this paper is the work of Hao et al. [2020] who consider sparse linear bandits. The authors utilize a "explore-then-commit" style algorithm to uniformly explore actions before using the Lasso to estimate the sparse linear parameter. We utilize a similar algorithm but allow for arbitrary interaction between neighboring units, instead using discrete Fourier analysis to linearly represent rewards [Negahban and Shah, 2012, O'Donnell, 2014, Agarwal et al., 2023]. This is similar to kernel bandits [Srinivas et al., 2009, Chowdhury and Gopalan, 2017, Whitehouse et al., 2024], which assume there exists a feature map such that the rewards can be linearly represented (non-sparsely) in a high-dimensional reproducing kernel Hilbert space. Also related are stochastic combinatorial bandits [Chen et al., 2013, Cesa-Bianchi and Lugosi, 2012], in which the action space is assumed to be a subset of $\{0, 1\}^N$ but rewards are typically inherently assumed to be linear in treatment assignments. That is, these works typically assume the reward $r = \langle \boldsymbol{\theta}, \mathbf{a} \rangle$ for $\mathbf{a} \in \{0, 1\}^N$, with valid actions $\mathbf{a}$ often having at most $s$ non-zero components. Our work (with $\mathcal{A} = 2$), considers an arbitrary function $r : \{0, 1\}^N \to \mathbb{R}$, but explicitly constructs a feature map via discrete Fourier analysis such that rewards can be represented linearly.

# 3 Model & Background

In this section, we first describe the problem setting, and our notion of regret. Then, we introduce the requisite background on discrete Fourier analysis that we will use to motivate our algorithm and theoretical analysis. Last, we introduce the model that we study in this paper. Throughout this paper, we use boldface to represent vectors and matrices.

## 3.1 Problem Set-up

We consider an agent that sequentially interacts with an environment consisting of $N$ individual units over a series of $T$ rounds. We index units $n \in [N]$, and rounds $t \in [T]$. At each time step $t$, the agent simultaneously administers each unit $n$ action (or treatment) $a \in [\mathcal{A}]$. Let $a_{nt}$ denote the treatment received by unit $n$ at time step $t$, and let $\mathbf{a}_t = (a_{1t}, \ldots, a_{Nt}) \in [\mathcal{A}]^N$ denote the entire treatment vector. Each unit $n$ possesses an unknown reward mapping $r_n : [\mathcal{A}]^N \to [0, 1]$. Note that we allow the reward for a given unit $n$ to depend on the treatments assigned to *all* other units, i.e., we allow for cross-unit *interference*. After assigning a treatment to all units in round $t$, the agent then observes the *noisy reward* for unit $n$ as $R_{nt} = r_n(\mathbf{a}_t) + \epsilon_{nt}$. Denote the vector of observed rewards as $\mathbf{R}_t := (R_{1t} \ldots R_{Nt})$. We assume the following standard condition on the noise $\epsilon_{nt}$.

**Assumption 1.** $(\epsilon_{nt} : n \in [N], t \in [T])$ *is a collection of mutually independent 1-sub-Gaussian random variables.*

---

[3]Specifically, it violates the stable unit treatment value assumption (SUTVA) [Rubin, 1978].

**Regret.** To measure the performance of the learning agent, we define the average reward function $\bar{r} : [\mathcal{A}]^N \to [0, 1]$ by $\bar{r}(\mathbf{a}) = \frac{1}{N} \sum_{n=1}^N r_n(\mathbf{a})$. Then, for a sequence of (potentially random) treatment assignments $\mathbf{a}_1 \ldots \mathbf{a}_T$, the regret at the horizon time $T$ is defined as the quantity

$$\text{Reg}_T = \sum_{t=1}^T \bar{r}(\mathbf{a}^*) - \sum_{t=1}^T \bar{r}(\mathbf{a}_t), \tag{1}$$

where $\mathbf{a}^* \in \arg\max_{\mathbf{a} \in [\mathcal{A}]^N} \bar{r}(\mathbf{a})$. In Sections 4 and 5, we provide and analyse algorithms that achieve small regret with high probability.

**Comparison to other works.** Previous works studying network bandits such as Jia et al. [2024] measure regret with respect to the best constant action $\mathbf{a}' := \arg\max_{a \in [\mathcal{A}]} \bar{r}(a\mathbf{1})$ where $\mathbf{1} \in \mathbb{R}^N$ denotes the all 1 vector of dimension $N$. We compare regret to the optimal action $\mathbf{a}^* \in \arg\max_{\mathbf{a} \in [\mathcal{A}]^N} \bar{r}(\mathbf{a})$, which is combinatorially more difficult to minimize since the policy space is exponentially larger ($\mathcal{A}^N$ vs $\mathcal{A}$). Our setup is also different than the traditional MAB setting since the agent in this problem does not observe a single scalar reward, but one for each unit (similar to semi-bandit feedback in the combinatorial bandits literature [Cesa-Bianchi and Lugosi, 2012]). As we show later, this crucially allows us to exploit local, unit-specific information that allow for better regret rates.

### 3.2 Background on Discrete Fourier Analysis

In this section, we provide background on discrete Fourier analysis, which we heavily employ in both our algorithm and analysis. Specifically, these Fourier-analytic tools provide a linear representation of the discrete unit-specific rewards $r_n : [\mathcal{A}]^N \to [0, 1]$, which will allow us to leverage well-studied linear bandit algorithms. For the rest of paper, assume $\mathcal{A}$ is a power of 2. If instead, if $2^\ell < \mathcal{A} < 2^{\ell+1}$ for some $\ell \geq 0$, we can redundantly encode actions to obtain $\mathcal{A}' = 2^{\ell+1}$ total treatments. As seen later, this encoding does not affect the overall regret.

**Boolean encoding of action space.** Since by assumption $\mathcal{A}$ is a power of 2, every action $a \in [\mathcal{A}]$ can be uniquely represented as a binary number using $\log_2(\mathcal{A})$ bits. Explicitly, let $\tilde{\mathbf{v}}(a) = (\tilde{v}_1(a), \ldots \tilde{v}_{\log_2(\mathcal{A})}(a)) \in \{0, 1\}^{\log_2(\mathcal{A})}$ denote this vectorized binary representation. For ease of analysis, we use the Boolean representation instead $\mathbf{v}(a) = 2\tilde{\mathbf{v}}(a) - \mathbf{1} \in \{-1, 1\}^{\log_2 \mathcal{A}}$. For $\mathbf{a} \in [\mathcal{A}]^N$, define $\mathbf{v}(\mathbf{a}) = (\mathbf{v}(a_1), \ldots, \mathbf{v}(a_N)) \in \{-1, 1\}^{N \log_2(\mathcal{A})}$. Note each action $\mathbf{a} \in [\mathcal{A}]^N$ corresponds to a unique Boolean vector $\mathbf{v}(\mathbf{a})$.

**Boolean representation of discrete functions.** Let $\mathcal{F} = \{f : [\mathcal{A}]^N \to \mathbb{R}\}$ and $\mathcal{F}_{\text{Bool}} = \{\tilde{f} : \{-1, 1\}^{N \log_2(\mathcal{A})} \to \mathbb{R}\}$ be the collection of all real-values functions defined on the set $[\mathcal{A}]^N$ and $\{-1, 1\}^{N \log_2(\mathcal{A})}$ respectively. Since every $\mathbf{a} \in [\mathcal{A}]^N$ has a uniquely Boolean representation $\mathbf{v}(\mathbf{a}) \in \{-1, 1\}^{N \log_2(\mathcal{A})}$, the set of functions $\mathcal{F}$ can be naturally identified within $\mathcal{F}_{\text{Bool}}$. Specifically, any $f \in \mathcal{F}$ can be identified with the function $\tilde{f} \in \mathcal{F}_{\text{Bool}}$ by $f(\cdot) = \tilde{f}(\mathbf{v}(\cdot))$.

**Fourier series of Boolean functions.** This identification is key for our use since the space of Boolean functions admits a number of attractive properties.

*(1) Hilbert space.* $\mathcal{F}_{\text{Bool}}$ forms a Hilbert space defined by the following inner product: for any $h, g \in \mathcal{F}_{\text{Bool}}$, $\langle h, g \rangle_B = \mathcal{A}^{-N} \sum_{\mathbf{x} \in \{-1,1\}^{N \log_2(\mathcal{A})}} h(\mathbf{x})g(\mathbf{x})$. This inner product induces the norm $\langle h, h \rangle_B := \|h\|_B^2 = \mathcal{A}^{-N} \sum_{\mathbf{x} \in \{-1,1\}^p} h^2(\mathbf{x})$.

*(2) Simple orthonormal basis.* For each subset $S \subset [N \log_2(\mathcal{A})]$, define a basis function $\chi_S(\mathbf{x}) = \prod_{i \in S} x_i$ where $x_i$ is the $i^{\text{th}}$ coefficient of $\mathbf{x} \in \{-1, 1\}^{N \log_2(\mathcal{A})}$. One can verify that for any $S \subset [N \log_2(\mathcal{A})]$, $\|\chi_S\|_B = 1$, and that $\langle \chi_S, \chi_{S'} \rangle_B = 0$ for any $S' \neq S$. Since $|\{\chi_S : S \subset [N \log_2(\mathcal{A})]\}| = \mathcal{A}^N$, the functions $\chi_S$ are an orthonormal basis of $\mathcal{F}_{\text{bool}}$. We refer to $\chi_S$ as the Fourier character for the subset $S$.

*(3) Linear Fourier expansion of $\mathcal{F}_{Bool}$.* Any $h \in \mathcal{F}_{\text{bool}}$ can be expanded via the following Fourier decomposition: $h(\mathbf{x}) = \sum_{S \subset [N \log_2(\mathcal{A})]} \theta_S \chi_S(\mathbf{x})$, where the Fourier coefficient $\theta_S$ is given by $\theta_S = \langle h, \chi_S \rangle_B$. For $h \in \mathcal{F}_{\text{Bool}}$, we refer to $\boldsymbol{\theta}_h = (\theta_S : S \subset [N \log_2(\mathcal{A})]) \in \mathbb{R}^{\mathcal{A}^N}$ as the vector of Fourier coefficients associated with it. For $\mathbf{x} \in \{-1, 1\}^{N \log_2(\mathcal{A})}$, let $\boldsymbol{\chi}(\mathbf{x}) = (\chi_S(\mathbf{x}) : S \in [N \log_2(\mathcal{A})]) \in$

$\{-1, 1\}^{\mathcal{A}^N}$ be the vector of associated Fourier character outputs. For $\mathbf{a} \in [\mathcal{A}]^N$, abbreviate $\chi_S(\mathbf{v}(\mathbf{a}))$ and $\boldsymbol{\chi}(\mathbf{v}(\mathbf{a}))$ as $\chi_S(\mathbf{a})$ and $\boldsymbol{\chi}(\mathbf{a})$ respectively.

### 3.3 Model: Sparse Network Interference

The unit-specific reward function $r_n : [\mathcal{A}]^N \to \mathbb{R}$ can be equivalently viewed as a real-valued Boolean function over the hypercube $\{-1, 1\}^{N \log_2(\mathcal{A})}$. That is, $r_n$ takes as input a vector of actions $\mathbf{a} \in [\mathcal{A}]^N$, converts it to a Boolean vector $\mathbf{v}(\mathbf{a}) \in \{-1, 1\}^{N \log_2(\mathcal{A})}$, and outputs a reward $r_n(\mathbf{a})$. From the discussion in Section 3.2, we can represent unit $n$'s reward as $r_n(\mathbf{a}) = \sum_{S \subset [N \log_2(\mathcal{A})]} \theta_{n,S} \chi_S(\mathbf{a}) = \langle \boldsymbol{\theta}_n, \boldsymbol{\chi}(\mathbf{a}) \rangle$, where $\boldsymbol{\theta}_n = (\theta_{n,S} : S \subseteq N \log_2(\mathcal{A})) \in \mathbb{R}^{\mathcal{A}^N}$ is a vector of Fourier coefficients.

Without any assumptions on the nature of the interference pattern, achieving low regret is impossible since it requires estimating $\mathcal{A}^N$ Fourier coefficients per unit. To overcome this fundamental challenge, we impose a natural structure on the interference pattern which assumes that the reward $r_n$ only depends on the the treatment assignment of a subset of $s$ units. This assumption is often observed in practice, e.g., the revenue of a good does not depend on discounts applied to all other goods, but only those applied to similar or related ones.

**Assumption 2.** *(Sparse Network Interference) For any unit $n \in [N]$, there exists a neighborhood $\mathcal{N}(n) \subset [N]$ of size $|\mathcal{N}(n)| \le s$ such that $r_n(\mathbf{a}) = r_n(\mathbf{b})$ for all $\mathbf{a}, \mathbf{b} \in \{-1, 1\}^{N \log_2 \mathcal{A}}$ satisfying $(a_m : m \in \mathcal{N}(n)) = (b_m : m \in \mathcal{N}(n))$.*

We typically assume that $n \in \mathcal{N}(n)$, i.e. unit $n$'s reward depends on its own treatment. This model allows for completely arbitrary interference between these $s$ units, generalizing the results of Jia et al. [2024] who allow for interaction between all $N$ units but assume the strength of interference decays with a particular notion of distance between units. Next, we show using our Fourier analytic tools, that Assumption 2 implies that the reward can be re-expressed as a sparse linear model. We prove the following in Appendix A.

**Proposition 3.1.** *Let Assumption 2 hold. Then, for any unit $n$, and action $\mathbf{a} \in [\mathcal{A}]^N$, we have the following representation of the reward $r_n(\mathbf{a}) = \langle \boldsymbol{\theta}_n, \boldsymbol{\chi}(\mathbf{a}) \rangle$, where $\|\boldsymbol{\theta}_n\|_0 \le \mathcal{A}^s$.*[4]

Proposition 3.1 shows sparse network interference implies $\boldsymbol{\theta}_n$ is $\mathcal{A}^s$ sparse with non-zero coordinates corresponding to the interactions of treatments between units in $\mathcal{N}(n)$. Indeed, the Boolean encoding $\mathbf{v}(a)$ can be represented as blocks of $\log_2(\mathcal{A})$ dimensional Boolean vectors:

$$\mathbf{v}(\mathbf{a}) = (\underbrace{\mathbf{v}(\mathbf{a})_{1:\log_2(\mathcal{A})}}_{\text{Unit 1's treatment}}, \ldots, \underbrace{\mathbf{v}(\mathbf{a})_{(i-1)\log_2(\mathcal{A})+1:i\log_2(\mathcal{A})}}_{\text{Unit } i\text{'s treatment}}, \ldots, \underbrace{\mathbf{v}(\mathbf{a})_{(N-1)\log_2(\mathcal{A})+1:N\log_2(\mathcal{A})}}_{\text{Unit } N\text{'s treatment}}).$$

Unit $n$'s reward depends on a small collection of these blocks, those indexed by its neighbors. Define

$$\mathcal{B}(n) := \left\{ i \in [N \log_2(\mathcal{A})] : i \in [(m-1)\log_2(\mathcal{A}) + 1 : m \log_2(\mathcal{A})] \text{ for some } m \in \mathcal{N}(n) \right\}.$$

$\mathcal{B}(n)$ contains the indices of $\mathbf{v}(a)$ corresponding to treatments of units $m \in \mathcal{N}(n)$ and the non-zero entries of $\boldsymbol{\theta}_n$ are indexed by subsets $S \subset \mathcal{B}(n)$. E.g., consider $N = 3$, $\mathcal{A} = 2$, with $\mathcal{N}(1) = \{1, 2\}$. Then $\mathcal{B}(1) = \{1, 2\}$ and $S \subset \mathcal{B}(n) = \{\emptyset, \{1\}, \{2\}, \{1, 2\}\}$, where $\emptyset$ is the empty set.

**Graphical interpretation.** Assumption 2 can be interpreted graphically as follows. Let $\mathcal{G} = ([N], \mathcal{E})$ denote a *directed* graph over the $N$ units, where $\mathcal{E} \subseteq [N] \times [N]$ denotes the edges of $\mathcal{G}$. For unit $n$, we add to the edge set $\mathcal{E}$ a directed edge $(n, m)$ for each $m \in \mathcal{N}(n)$, thus justifying calling $\mathcal{N}(n)$ the *neighborhood* of $n$. That is, unit $n$'s reward is affected by the treatment of another unit $m$ only if there is a directed edge from $n$ to $m$. See Figure 1 for an example of a network graph $\mathcal{G}$.

## 4 Network Multi-Armed Bandits with Known Interference

We now present our algorithms and regret bounds when the interference pattern is known, i.e. the learner observes $\mathcal{G}$ and knows $\mathcal{N}(n)$ for each unit $n$. The unknown case is analysed in Section 5. Assuming knowledge of $\mathcal{G}$ is reasonable in e-commerce, where the platform (learner) assigning discounts (treatments) to goods (units) understands the underlying similarity between goods.

---

[4] For a vector $\mathbf{x} \in \mathbb{R}^d$, we define $\|x\|_0 := \sum_{i=1}^d \mathbb{1}(x_i \ne 0)$

Our algorithm requires the following additional notation: for $\mathbf{a} \in [\mathcal{A}]^N$, let $\boldsymbol{\chi}^{\mathbf{a}}(\mathcal{B}_n) = (\chi_S(\mathbf{a}) : S \subset \mathcal{B}(n)) \in \{-1,1\}^{\mathcal{A}^s}$, where $\chi_S(\mathbf{a})$ are the Fourier characteristics corresponding to subsets of $\mathcal{B}(n)$. Further, let $\mathcal{U}([\mathcal{A}]^N)$ denote the uniform discrete distribution on the action space $[\mathcal{A}]^N$.

---

**Algorithm 1** Network Explore-Then-Commit with Known Interference

---

1: **Input:** Time horizon $T$, exploration steps $E$, interference graph $\mathcal{G} = ([N], \mathcal{E})$.
2: Sample $\mathbf{a}_1, \ldots, \mathbf{a}_E \sim_{\text{i.i.d.}} \mathcal{U}\left([\mathcal{A}]^N\right)$
3: Observe reward vectors $\mathbf{R}_t = (R_{1t}, \cdots, R_{Nt})$ for $t \in [E]$, where $R_{nt} = \langle \boldsymbol{\theta}_n, \boldsymbol{\chi}(\mathbf{a}_t) \rangle + \epsilon_{nt}$.
4: **for** $n \in [N]$ **do**
5:      Let $\mathbf{X}_n = (\boldsymbol{\chi}^{\mathbf{a}_i}(\mathcal{B}_n) : i \in [E]) \in \{-1,1\}^{E \times \mathcal{A}^s}$ // $\boldsymbol{\chi}^{\mathbf{a}}(\mathcal{B}_n) = (\chi_S(\mathbf{a}) : S \subset \mathcal{B}(n))$
6:      Let $\mathbf{Y}_n = (R_{n1}, \ldots, R_{nE})$.
7:      Set $\widetilde{\boldsymbol{\theta}}_n := \arg\min_{\boldsymbol{\theta} \in \mathbb{R}^{\mathcal{A}^s}} \| \mathbf{Y}_n - \mathbf{X}_n \boldsymbol{\theta} \|_2^2$
8:      Define $\widehat{\boldsymbol{\theta}}_n$ by $\widehat{\theta}_{nS} = \widetilde{\theta}_{nS}$ if $S \subset \mathcal{B}(n)$ else set $\widehat{\theta}_{nS} = 0$. // Coordinates of $\widetilde{\boldsymbol{\theta}}_n$ indexed by subsets of $\mathcal{B}(n)$
9: Set $\widehat{\boldsymbol{\theta}} := N^{-1} \sum_{n=1}^{N} \widehat{\boldsymbol{\theta}}_n$.
10: Play $\widehat{\mathbf{a}} := \arg\max_{\mathbf{a} \in [\mathcal{A}]^N} \langle \widehat{\boldsymbol{\theta}}, \boldsymbol{\chi}(\mathbf{a}) \rangle$ for the $T - E$ remaining rounds.

---

Algorithm 1 is a "explore-then-commit" style which operates in two phases. First, the learner assigns units treatments uniformly at random for $E$ rounds, and observes rewards for each unit. In the second phase, the algorithm performs least squares regressions of the observed rewards against $\boldsymbol{\chi}^{\mathbf{a}}(\mathcal{B}_n)$ for each unit $n$. This is because when $\mathcal{G}$ is known, the learner knows the positions of the non-zero elements of $\boldsymbol{\theta}_n$ which are precisely the subsets of $\mathcal{B}(n)$. Once the estimates $\widehat{\boldsymbol{\theta}}_n$ are obtained for each unit, they are aggregated to estimate the average reward for each action $\mathbf{a} \in [\mathcal{A}]^N$. In the remaining $T - E$ rounds, the learner greedily plays the action with the highest estimated average reward.

**Determining exploration length $E$.** Theoretically, we detail the length of $E$ below to achieve low regret in Theorem 4.1. Practically, the learner can continue to explore and assess the error of the learnt $\widehat{\boldsymbol{\theta}}_n$ via cross-validation (CV). Once the CV error for all units falls below a (user-specified) threshold, commit to the action with highest average reward. We use this approach for selecting $E$ in our simulations in Section 6.

### 4.1 Regret Analysis

Here, we establish high-probability regret bounds of Algorithm 1 using $O(\cdot)$ notation. We prove the following in Appendix B.

**Theorem 4.1.** *Suppose Assumptions 1 and 2 hold. For $T = \Omega\left(A^{2s}[\log(2N/\delta) + s\log(\mathcal{A})]\right)$ and any failure probability $\delta \in (0,1)$, Algorithm 1 run with $E := (T\mathcal{A}^s)^{2/3}\left[\log\left(\frac{N}{\delta}\right) + s\log(\mathcal{A})\right]^{1/3}$ satisfies*

$$\text{Reg}_T = O\left([s\log(\mathcal{A}/\delta)]^{1/3}(T\mathcal{A}^s)^{2/3}\right),$$

*with probability at least $1 - \delta$.*

Establishing Theorem 4.1 requires trading-off the exploration time $E$ to accurately estimate $\boldsymbol{\theta}_n$ with the exploitation time. It also requires $T$ to be large enough such that we can accurately estimate $\boldsymbol{\theta}_n$. Next, we compare regret of Algorithm 1 to other methods, ignoring any dependencies on logarithmic factors to ease the discussion.

**Comparison to other approaches.**

(a) *Naïve MAB learner.* A naïve learner who treats the entire network of units as a single multi-armed bandit system with $\mathcal{A}^N$ actions will obtain regret $\widetilde{O}(\sqrt{T\mathcal{A}^N})$. For sparse networks with $s \ll N$ and $T \ll \mathcal{A}^N$, our regret bound is significantly tighter.

(b) *Global estimation.* An alternate algorithm would be to estimate Fourier coefficients $\boldsymbol{\theta} := 1/N \sum_{i=1}^{N} \boldsymbol{\theta}_n$ of $\bar{r}$ directly rather than estimate each $\boldsymbol{\theta}_n$ (i.e., $r_n$) individually. That is, perform

the least squares regression by compressing the observed, unit-specific rewards into $\overline{R}_t :=$ $N^{-1}\sum_{n=1}^{N} R_{tn}$. An analysis similar to the one presented in Appendix B would yield rate of $\widetilde{O}(s^{1/3}(TN\mathcal{A}^s)^{2/3})$, which suffers an additional $N^{2/3}$ cost as compared to Theorem 4.1.

(c) *Jia et al. [2024].* Comparing regret to this work is difficult because they assume decaying interference strength on a grid-like network structure and establish regret only with respect to the best constant action, i.e., $\mathbf{a}' := \arg\max_{a\in[\mathcal{A}]} \bar{r}(a\mathbf{1})$. We also note that the framework of Jia et al. [2024] is closer to that of adversarial bandits, whereas our framework is closer to that of stochastic bandits.

# 5 Network Multi-Armed Bandits with Unknown Interference

Next, we consider the case in which the underlying network $\mathcal{G}$ governing interference is not known. We present Algorithm 2, which extends Algorithm 1 to account for the fact that the learner does not observe the network graph $\mathcal{G}$ and thus does not know $\mathcal{N}(n)$ for all $n$. Unknown network interference is common in medical trials, e.g., vaccine roll-outs where an individual's social network (i.e., $\mathcal{G}$) is unavailable to the learner.

---

**Algorithm 2** Network Explore-Then-Commit with Unknown Interference

---

1: **Input:** Time horizon $T$, exploration steps $E$, regularization parameter $\lambda > 0$
2: Sample $\mathbf{a}_1, \ldots, \mathbf{a}_E \sim_{\text{i.i.d.}} \mathcal{U}\left([\mathcal{A}]^N\right)$
3: Observe reward vectors $\mathbf{R}_t = (R_{1t}, \cdots, R_{Nt})$ for $t \in [E]$, where $R_{nt} = \langle \boldsymbol{\theta}_n, \boldsymbol{\chi}(\mathbf{a}_t) \rangle + \epsilon_{nt}$.
4: Let $\mathbf{X} = (\boldsymbol{\chi}(\mathbf{a}_i) : i \in [E]) \in \{-1, 1\}^{E \times \mathcal{A}^N}$
5: **for** $n \in [N]$ **do**
6:     Let $\mathbf{Y}_n := (R_{n1}, \ldots, R_{nE})$.
7:     Set $\widehat{\boldsymbol{\theta}}_n := \arg\min_{\boldsymbol{\theta} \in \mathbb{R}^{\mathcal{A}^N}} \left\{ \frac{1}{2E}\|\mathbf{X}\boldsymbol{\theta} - \mathbf{Y}_n\|_2^2 + \lambda\|\boldsymbol{\theta}\|_1 \right\}$
8: Set $\widehat{\boldsymbol{\theta}} := N^{-1}\sum_{n=1}^{N} \widehat{\boldsymbol{\theta}}_n$.
9: Play $\widehat{\mathbf{a}} := \arg\max_{\mathbf{a} \in [\mathcal{A}]^N} \langle \widehat{\boldsymbol{\theta}}, \boldsymbol{\chi}(\mathbf{a}) \rangle$ for the $T - E$ remaining rounds.

---

Algorithm 2 is similar to Algorithm 1, but differs in how it learns $\boldsymbol{\theta}_n$. Since $\mathcal{G}$ is unknown, the learner cannot identify the Fourier characteristics which correspond to the non-zero elements of $\boldsymbol{\theta}_n$. Therefore, we regress against the entire Fourier characteristic $\boldsymbol{\chi}(\mathbf{a})$, using Lasso instead of ordinary least squares to adapt to the underlying sparsity of $\boldsymbol{\theta}_n$. A similar CV approach, as discussed after Algorithm 1, can be used to determine both the exploration length $E$, and regularization parameter $\lambda$.

**Low-order interactions.** When $\mathcal{A}^N$ is very large, the computational cost of running the Lasso can be large. Further, if the underlying network is indeed believed to be sparse, one can regress against all characteristics $\chi_S$ where $|S| \leq d$. A similar approach is explored in Yu et al. [2022]. In practice, one can choose degree $d$ via CV.

**Partially observed network graph $\mathcal{G}$.** In many settings, network interference graphs $\mathcal{G}$ are partially observed. For example, on e-commerce platforms, interference patterns between established classes of goods is well-understood, but might be less so for newer products. Our framework can naturally be adapted to this setting by running Algorithm 1 on the observed portion of $\mathcal{G}$, and Algorithm 2 on the unobserved graph. Specifically, if $\mathcal{N}(n)$ is observed for unit $n$, replace the Lasso in line 7 of Algorithm 2 with OLS (i.e., line 8) in Algorithm 1.

## 5.1 Regret Analysis

We now establish high-probability bounds on the regret for Algorithm 2 in Theorem 5.1. We prove the following in Appendix C.

**Theorem 5.1.** *Suppose Assumptions 1 and 2 hold, and assume $T = \Omega(A^{2s}\left[\log(N/\delta) + N\log(\mathcal{A})\right])$. Then, with failure probability $\delta \in (0, 1)$, Algorithm 2 run with $\lambda = 4\sqrt{E^{-1}\log(2\mathcal{A}^N)} + 4\sqrt{E^{-1}\log\left(\frac{2N}{\delta}\right)}$ where $E := (T\mathcal{A}^s)^{2/3}\left[\log\left(\frac{N}{\delta}\right) + N\log(\mathcal{A})\right]^{1/3}$ satisfies*

$$\text{Reg}_T = O\left(\left[N\log(\mathcal{A}/\delta)\right]^{1/3}\left(T\mathcal{A}^s\right)^{2/3}\right)$$

*with probability at least $1 - \delta$.*

We note the regret bound requires the horizon $T$ to be sufficiently large in order to learn the network graph $\mathcal{G}$ — a necessary detail in order to ensure Lasso convergence. This is because the proof of Theorem 5.1 requires establishing that the matrix of Fourier coefficients for the sampled actions (i.e., design matrix $\mathbf{X}$) satisfies the the necessary regularity conditions to learn $\boldsymbol{\theta}_n$ accurately. Specifically, we show that $\mathbf{X}$ is incoherent, i.e., approximately orthogonal, with high probability. See Appendix C for a formal definition of incoherence, and Rigollet and Hütter [2023], Wainwright [2019] for a detailed study of the Lasso.

**Comparison to other approaches.** Algorithm 2 achieves the same dependence in $\mathcal{A}, s, T$ as in the known interference case, but pays a factor of $N^{1/3}$ as compared to $s^{1/3}$. This additional cost which is logarithmic in the ambient dimension $\mathcal{A}^N$ is typical in sparse online learning. This regret rate is still significantly lower than naïve approaches that scale as $O(\sqrt{\mathcal{A}^N T})$ when one assumes $T$ is much smaller then $\mathcal{A}^N$. Further, as argued before, estimating per-unit rewards (i.e., $\boldsymbol{\theta}_n$) results in lower regret as compared to directly estimating $\bar{r}$ by a factor of $N^{2/3}$.

**Dependence on horizon $T$.** Generally, the dependence on $T$ cannot be improved. Hao et al. [2020] lower bound regret for sparse linear bandits as $\widetilde{\Omega}((\text{sparsity} \cdot T)^{2/3})$, i.e., $\widetilde{\Omega}((\mathcal{A}^s \cdot T)^{2/3})$ in our setting. They show improved dependence on $T$ can only be achieved under stronger assumptions on the size of non-zero coefficients of $\boldsymbol{\theta}_n$.

# 6 Simulations

In this section, we perform simulations to empirically validate our algorithms and theoretical findings. We compare Algorithms 1 and 2 to UCB. We could not compare to Jia et al. [2024] since we did not find a public implementation. For our Algorithms, we choose all hyper-parameters via 3-fold CV, and use the `scikit-learn` implementation of the Lasso. Code for our methods and experiments can be found at `https://github.com/aagarwal1996/NetworkMAB`. Our experimental setup and results are described below.

**Data Generating Process.** We generate interference patterns with varying number of units $N \in \{5, \ldots, 10\}$, and $\mathcal{A} = 2$. For each $N$, we use $s = 4$. We generate rewards $r_n = \langle \boldsymbol{\theta}_n, \boldsymbol{\chi}(\mathbf{a}) \rangle$, where the non-zero elements of $\boldsymbol{\theta}_n$ (i.e., $\theta_{n,S}$ for $S \subset \mathcal{B}_n$) are drawn uniform from $[0, 1]$. We normalize rewards so that they are contained in $[0, 1]$, and add 1 sub-gaussian noise to sampled rewards. We measure regret as we vary $T$, and set a max horizon of $T_{\max} = 10 \cdot 2^N$ for each $N$. Classical MAB algorithms need the horizon $T$ to satisfy $T > 2^N$ since they first explore by pulling all $2^N$ arms. We emphasize that these time horizons scaling as $T = C \cdot \mathcal{A}^N$ are often unreasonable in practice, as even for $\mathcal{A} = 2$ and $N = 100$ there would already be $\approx 1.27\mathrm{e}^{30}$ actions to explore. We include such large time horizons for the sake of making a complete comparison. Our methods circumvent the need for exponentially large exploration times by effectively exploiting sparsity.

**Results.** We plot the regret at the maximum horizon time as a function of $N$, and the cumulative regret as we vary $T$ for $N = 13$ in Figure 2 below. Our results are averaged over 5 repetitions, with shaded regions representing 1 standard deviation measured across repetitions. Algorithms 1 and 2 are denoted by Network MAB (Known) and Network MAB (Unknown) respectively. We discuss both sets of plot separately below.

*Regret Scaling with $N$.* We plot the cumulative regret when $T = T_{\max}$ for $N = 9$ in Figure 2 (a). Classical MAB algorithms such as UCB see an exponential growth in the regret as $N$ increases. Both Algorithm 1 and Algorithm 2 have much milder scaling with $N$. Algorithm 1 uses $\mathcal{G}$ to reduce the ambient dimension of the regression, hence suffering less dependence on $N$ as compared to Algorithm 2.

*Regret Scaling with $T$.* We plot the cumulative regret for $N = 9$ in Figure 2 (b). Despite the poorer scaling of our regret bounds with $T$, our algorithms lead to significantly better regret than UCB which takes a large horizon to converge. Algorithm 1 is able to end its exploration phase earlier than algorithm 2 since it does not need additional samples to learn the sparsity unlike the Lasso.

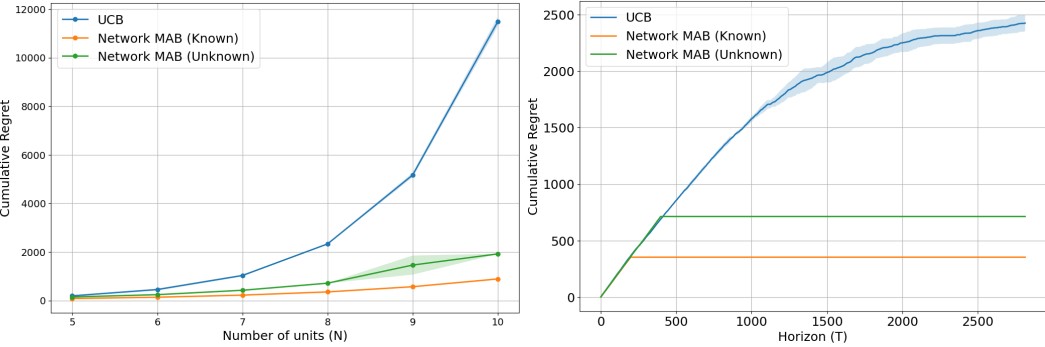

(a) Cumulative regret vs number of units $N$.   (b) Cumulative regret scaling vs horizon $T$.

**Figure 2:** We simulate rewards via a sparse network interference pattern, and plot the cumulative regret as a function of $N$ and $T$. Our Network MAB algorithms out-perform UCB, irrespective of knowledge of $\mathcal{G}$, and does not suffer exponential dependence in number of units $N$. The results also confirm our theoretical results that knowledge of $\mathcal{G}$ leads Algorithm 1 to have milder dependence in $N$ and better regret than Algorithm 2.

## 7 Conclusion

This paper introduces a framework for regret minimization in MABs with network interference, a ubiquitous problem in practice. We study this problem under a natural sparsity assumption on the interference pattern and provide simple algorithms both when the network graph is known and unknown. Our analysis establishes low regret for these algorithms and numerical simulations corroborate our theoretical findings. The results in this paper also significantly generalize previous works on MABs with network interference by allowing for arbitrary and unknown (neighbourhood) interference, as well as comparing to a combinatorially more difficult optimal policy. This paper also suggests future directions for research such as designing algorithms that achieve better dependence on $T$ in the known graph setting. Establishing lower bounds to understand optimal algorithms will also be valuable future work. Further extensions could also include considering interference in contextual bandits or reinforcement learning problems. We also hope this work serves as a bridge between online learning and discrete Fourier analysis.

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

# A  Proof of Proposition 3.1

By the discussion in Section 3, recall that for any action $\mathbf{a} \in [\mathcal{A}]^N$ and unit $n$, the reward can be expressed as $r_n = \langle \boldsymbol{\theta}_n, \boldsymbol{\chi}(\mathbf{a}) \rangle$. To establish the proof, it suffices to show that for any $S \subset [N \log_2(\mathcal{A})]$ satisfying $S \setminus \mathcal{B}(n) \neq \emptyset$, $\langle \chi_S, r_n \rangle_B = 0$. Let $i \in S \setminus \mathcal{B}(n)$ be an arbitrary index, then, we have,

$$\langle \chi_S, r_n \rangle_B = \mathcal{A}^{-N} \sum_{\mathbf{x} \in \{-1,1\}^{N \log_2(\mathcal{A})}} r_n(\mathbf{x}) \chi_S(\mathbf{x})$$

$$= \mathcal{A}^{-N} \left\{ \sum_{\substack{\mathbf{x} \in \{-1,1\}^{N \log_2(\mathcal{A})} \\ x_i = 1}} r_n(\mathbf{x}) \chi_S(\mathbf{x}) + \sum_{\substack{\mathbf{x} \in \{-1,1\}^{N \log_2(\mathcal{A})} \\ x_i = -1}} r_n(\mathbf{x}) \chi_S(\mathbf{x}) \right\}$$

$$= \mathcal{A}^{-N} \left\{ \sum_{\substack{\mathbf{x} \in \{-1,1\}^{N \log_2(\mathcal{A})} \\ x_i = 1}} x_i r_n(\mathbf{x}) \chi_{S \setminus \{i\}}(\mathbf{x}) + \sum_{\substack{\mathbf{x} \in \{-1,1\}^{N \log_2(\mathcal{A})} \\ x_i = -1}} x_i r_n(\mathbf{x}) \chi_{S \setminus \{i\}}(\mathbf{x}) \right\}$$

$$= \mathcal{A}^{-N} \left\{ \sum_{\substack{\mathbf{x} \in \{-1,1\}^{N \log_2(\mathcal{A})} \\ x_i = 1}} r_n(\mathbf{x}) \chi_{S \setminus \{i\}}(\mathbf{x}) - \sum_{\substack{\mathbf{x} \in \{-1,1\}^{N \log_2(\mathcal{A})} \\ x_i = -1}} r_n(\mathbf{x}) \chi_{S \setminus \{i\}}(\mathbf{x}) \right\}$$

$$= 0,$$

where the final equality follows from the fact that, by Assumption 2, $r_n(\mathbf{x}) = r_n(\mathbf{x}')$ when $\mathbf{x}$ and $\mathbf{x}'$ differ only in positions indexed by $i \notin \mathcal{B}(n)$. Thus, the only subsets $S \subset [N \log_2(\mathcal{A})]$ where we can have $\langle r_n, \chi_S \rangle_B \neq 0$ are those satisfying $S \subset \mathcal{B}(n)$, which proves the desired result.

# B  Proofs for for Known Interference

In this section, we prove Theorem 4.1. We establish helper lemmas before proving Theorem 4.1.

## B.1  Helper Lemmas

Recall the following notation before establishing our results. We defined $\mathcal{B}(n) := \{i \in [N \log_2(\mathcal{A})] : i \in [(m-1) \log_2(\mathcal{A}) + 1 : m \log_2(\mathcal{A})]$ for $m \in \mathcal{N}(n)\}$ as the set of indices of the treatment vector $\mathbf{v}(a) \in \{-1, 1\}^{N \log_2(\mathcal{A})}$ belonging to neighbors $m \in \mathcal{N}(n)$. Additionally, $\mathbf{X}_n = (\boldsymbol{\chi}^{\mathbf{a}_i}(\mathcal{B}_n) : i \in [E]) \in \{-1, 1\}^{E \times \mathcal{A}^s}$, where $\boldsymbol{\chi}^{\mathbf{a}}(\mathcal{B}_n) = (\chi_S(\mathbf{a}) : S \subset \mathcal{B}(n)) \in \{-1, 1\}^{\mathcal{A}^s}$. For a matrix $\mathbf{A} \in \mathbb{R}^{N \times d}$, let $\sigma_{\min}(\mathbf{A})$ denote its minimum singular value. To proceed, we quote the following theorem.

**Lemma B.1** (Theorem 5.41 in Vershynin [2018]). *Let $\mathbf{A} \in \mathbb{R}^{N \times d}$ such that its rows $\mathbf{A}_i$ are independent isotropic random vectors in $\mathbb{R}^d$. If $\|\mathbf{A}_i\|_2 \leq \sqrt{m}$ almost surely for all $i \in [N]$, then, with probability at least $1 - \delta$, one has*

$$\sigma_{\min}(\mathbf{A}) \geq \sqrt{N} - \sqrt{cm \log(2d/\delta)}$$

*for universal constant $c > 0$.*

**Lemma B.2** (Minimum Eigenvalue of Fourier Characteristics). *There exists a positive constant $C_4 > 0$ such that if $E \geq C_4 \mathcal{A}^s \log(2\mathcal{A}^s/\delta)$, then,*

$$\sigma_{\min}\left(\frac{\mathbf{X}_n^T \mathbf{X}_n}{E}\right) \geq \frac{1}{2},$$

*with probability at least $1 - \delta$.*

*Proof.* We begin by showing the conditions for Lemma B.1 are satisfied. First, we prove $\mathbf{X}_n$ is isotropic, i.e., $\mathbb{E}[\boldsymbol{\chi}^{\mathbf{a}}(\mathcal{B}_n)(\boldsymbol{\chi}^{\mathbf{a}}(\mathcal{B}_n))^T] = \mathbf{I}_{\mathcal{A}^s}$, where the expectation is taken over uniformly

sampling actions $\mathbf{a}$ uniformly and random from $[\mathcal{A}]^N$. This follows since for any two subsets $S, S' \subset [N \log_2(\mathcal{A})]$,

$$\mathbb{E}[\chi_S(\mathbf{a})\chi_{S'}(\mathbf{a})] = \frac{1}{\mathcal{A}^N} \sum_{\mathbf{a} \in \mathcal{A}^N} \chi_S(\mathbf{a})\chi_{S'}(\mathbf{a})$$

$$= \langle \chi_S, \chi_{S'} \rangle_B = \mathbb{1}[S = S']$$

Since, $\boldsymbol{\chi}^{\mathbf{a}}(\mathcal{B}_n) \in \{-1,1\}^{\mathcal{A}^s}$ for all actions $\mathbf{a} \in [\mathcal{A}]^N$, $\|\boldsymbol{\chi}^{\mathbf{a}}(\mathcal{B}_n)\|_2 \leq \sqrt{\mathcal{A}^s}$. Hence, by Lemma B.1, $\sigma_{\min}(\mathbf{X}_n) \geq \sqrt{E} - \sqrt{c \log(2\mathcal{A}^s/\delta)\mathcal{A}^s}$. Next, using the fact that $\sigma_{\min}(\mathbf{X}_n^T\mathbf{X}_n) = \sigma_{\min}^2(\mathbf{X}_n)$, we get that

$$\sigma_{\min}\left(\frac{\mathbf{X}_n^T\mathbf{X}_n}{E}\right) \geq \frac{E - 2\sqrt{cE\mathcal{A}^s \log(2\mathcal{A}^s/\delta)}}{E}.$$

Finally, plugging in $\mathcal{A}^s \log(2\mathcal{A}^s/\delta) \leq E/C$ for an appropriate $C$ gives us the claimed result. $\square$

We quote the following theorem regarding the $\|\cdot\|_2$ error of $\widehat{\boldsymbol{\theta}}_n$.

**Lemma B.3.** *[Theorem 2.2 in Rigollet and Hütter [2023]] Assume that $\mathbf{Y} = \mathbf{X}\boldsymbol{\theta}^* + \epsilon$, where $\epsilon$ is 1 sub-Gaussian, where $\mathbf{X} \in \mathbb{R}^{E \times d}$. If $d \leq E$, and covariance matrix $\boldsymbol{\Sigma}_X = (\mathbf{X}^T\mathbf{X})/E$ has rank $d$, then we have with probability at least $1 - \delta$,*

$$\|\mathbf{X}\boldsymbol{\theta}^* - \mathbf{X}\widehat{\boldsymbol{\theta}}\|_2 \leq C_1 \sqrt{\frac{d + \log(1/\delta)}{E}},$$

*where $\widehat{\boldsymbol{\theta}} = \arg\min_{\boldsymbol{\theta} \in \mathbb{R}^d} \|\mathbf{Y} - \mathbf{X}\boldsymbol{\theta}\|_2^2$ is the least squares estimator, and $C_1 > 0$ is a positive universal constant.*

While the above lemma bounds the mean-squared error the least-squares estimate, in our applications we can about bounding the $\ell_2$ distance between $\boldsymbol{\theta}^*$ and $\widehat{\boldsymbol{\theta}}$. Simple rearrangement on the above implies that, with probability at least $1 - \delta$, we actually have

$$\|\boldsymbol{\theta}^* - \widehat{\boldsymbol{\theta}}\|_2 \leq C_1 \sqrt{\frac{d + \log(1/\delta)}{E \cdot \sigma_{\min}(\boldsymbol{\Sigma}_X)}}.$$

If, in particular, $\sigma_{\min}\left(\frac{\mathbf{X}^\top\mathbf{X}}{E}\right) \geq 1/2$, the above can be simplified to

$$\|\boldsymbol{\theta}^* - \widehat{\boldsymbol{\theta}}\|_2 \leq C_2 \sqrt{\frac{d + \log(1/\delta)}{E}}$$

with probability at least $1 - \delta$ for some new, appropriate universal constant $C_2 > 0$.

## B.2 Proof of Theorem 4.1

*Proof.* Recall the notation $\widehat{\boldsymbol{\theta}} = N^{-1} \sum_{n=1}^N \widehat{\boldsymbol{\theta}}_n$, and $\widehat{\mathbf{a}} = \arg\max_{\mathbf{a} \in [\mathcal{A}]^N} \langle \widehat{\boldsymbol{\theta}}, \boldsymbol{\chi}(\mathbf{a}) \rangle$. The average reward $\overline{r}(\widehat{\mathbf{a}})$ can be bounded using the definition of $\widehat{\mathbf{a}}$ and Holder's inequality as follows,

$$\overline{r}(\mathbf{a}^*) - \overline{r}(\widehat{\mathbf{a}}) = \langle \boldsymbol{\theta}, \boldsymbol{\chi}(\mathbf{a}^*) - \boldsymbol{\chi}(\widehat{\mathbf{a}}) \rangle$$

$$= \langle \boldsymbol{\theta} - \widehat{\boldsymbol{\theta}}, \boldsymbol{\chi}(\mathbf{a}^*) - \boldsymbol{\chi}(\widehat{\mathbf{a}}) \rangle + \underbrace{\langle \widehat{\boldsymbol{\theta}}, \boldsymbol{\chi}(\mathbf{a}^*) - \boldsymbol{\chi}(\widehat{\mathbf{a}}) \rangle}_{\leq 0}$$

$$\leq \langle \boldsymbol{\theta} - \widehat{\boldsymbol{\theta}}, \boldsymbol{\chi}(\mathbf{a}^*) - \boldsymbol{\chi}(\widehat{\mathbf{a}}) \rangle$$

$$= \frac{1}{N} \sum_{i=1}^N \langle \boldsymbol{\theta}_n - \widehat{\boldsymbol{\theta}}_n, \boldsymbol{\chi}(\mathbf{a}^*) - \boldsymbol{\chi}(\widehat{\mathbf{a}}) \rangle$$

$$= \frac{1}{N} \sum_{i=1}^N \langle \boldsymbol{\theta}_n - \widehat{\boldsymbol{\theta}}_n, \boldsymbol{\chi}^{\mathbf{a}^*}(\mathcal{B}_n) - \boldsymbol{\chi}^{\widehat{\mathbf{a}}}(\mathcal{B}_n) \rangle$$

$$\leq \frac{1}{N} \sum_{i=1}^N \|\boldsymbol{\theta}_n - \widehat{\boldsymbol{\theta}}_n\|_2 \|\boldsymbol{\chi}^{\mathbf{a}^*}(\mathcal{B}_n) - \boldsymbol{\chi}^{\widehat{\mathbf{a}}}(\mathcal{B}_n)\|_2$$

Using $\|\chi^{\mathbf{a}}(\mathcal{B}_n)\|_2 \leq \sqrt{\mathcal{A}^s}$ then gives us

$$\bar{r}(\mathbf{a}^*) - \bar{r}(\widehat{\mathbf{a}}) \leq \frac{\sqrt{\mathcal{A}^s}}{N} \sum_{i=1}^{N} \|\boldsymbol{\theta}_n - \widehat{\boldsymbol{\theta}}_n\|_2 \qquad (2)$$

Next, define "good" events for any unit $n \in [N]$ as

$$G_{n1} := \left\{ \sigma_{\min}\left(\frac{\mathbf{X}_n^T \mathbf{X}_n}{E}\right) \geq \frac{1}{2} \right\} \quad \text{and} \quad G_{n2} := \left\{ \|\widehat{\boldsymbol{\theta}}_n - \boldsymbol{\theta}_n\|_2 \leq C_2 \sqrt{E^{-1}\left[\mathcal{A}^s + \log\left(\frac{4N\mathcal{A}^s}{\delta}\right)\right]} \right\},$$

where $C_2 > 0$ is as stated above. Notice that there exists a sufficiently large universal constant $C_3 > 0$, such that $T \geq C_3\left(A^{2s}[\log(2N/\delta) + s\log(\mathcal{A})]\right)$ implies $E = (T\mathcal{A}^s)^{2/3}\left[\log\left(\frac{N}{\delta}\right) + s\log(\mathcal{A})\right]^{1/3} \geq C_4\mathcal{A}^s\log(4N\mathcal{A}^s/\delta)$. Hence, for any given $n \in [N]$, we have via Lemma B.2 that $\mathcal{G}_{n1}$ holds with probability $1 - \frac{\delta}{2N}$. Conditioned on $\mathcal{G}_{n1}$, we get that $\mathbb{P}(\mathcal{G}_{n2}|\mathcal{G}_1) \geq 1 - \frac{\delta}{2N}$. Summarizing, we get that for any $n \in [N]$, the following holds

$$\|\widehat{\boldsymbol{\theta}}_n - \boldsymbol{\theta}_n\|_2 \leq C_2 \sqrt{E^{-1}\left[\mathcal{A}^s + \log\left(\frac{4N\mathcal{A}^s}{\delta}\right)\right]} \leq C_5 \sqrt{E^{-1}\mathcal{A}^s \log\left(\frac{4N\mathcal{A}^s}{\delta}\right)},$$

with probability at least $\left(1 - \frac{\delta}{2N}\right)^2 \geq 1 - \delta/N$, where $C_5 > 0$ is an appropriate constant. Taking a union bound over all $N$ units, and then substituting into (2) gives us

$$\bar{r}(\mathbf{a}^*) - \bar{r}(\widehat{\mathbf{a}}) \leq C_5 \mathcal{A}^s \sqrt{E^{-1} \log\left(\frac{4N\mathcal{A}^s}{\delta}\right)}$$

Finally, using this, the cumulative regret can be upper bounded with probability $1 - \delta$ as follows:

$$\begin{aligned}
\text{Reg}_T &= \sum_{t=1}^{T} \left(\bar{r}(\mathbf{a}^*) - \bar{r}(\widehat{\mathbf{a}})\right) \\
&= \sum_{t=1}^{E} \left(\bar{r}(\mathbf{a}^*) - \bar{r}(\widehat{\mathbf{a}})\right) + \sum_{t=E+1}^{T} \left(\bar{r}(\mathbf{a}^*) - \bar{r}(\widehat{\mathbf{a}})\right) \\
&\leq E + C_5 T \mathcal{A}^s \sqrt{E^{-1} \log\left(\frac{4N\mathcal{A}^s}{\delta}\right)}
\end{aligned}$$

Substituting $E$ as in the theorem statement completes the proof. $\qquad\square$

## C Proofs for Unknown Interference

In this appendix, we prove Theorem 5.1. Our proof requires the following lemmas.

### C.1 Helper Lemmas for Theorem 5.1

The first lemma we prove details the (high-probability) incoherence guarantees of the uniformly random design matrix under the Fourier basis. Recall the following notation before stating and proving our results. We denote $E$ as our exploration length, and $\chi(\mathbf{a}_t)$ as the Fourier characteristic associated with action $\mathbf{a}_t \in [\mathcal{A}]^N$. Let $\mathbf{X} = (\chi(\mathbf{a}_t) : t \in [E]) \in \{-1, 1\}^{E \times \mathcal{A}^N}$. Additionally, we require the following definition of incoherence.

**Definition C.1.** *We say a matrix $\mathbf{A} \in \mathbb{R}^{E \times d}$ is $s$-incoherent if $\|\mathbf{A}^\top \mathbf{A} - \mathbf{I}_d\|_\infty \leq \frac{1}{32s}$, where $\mathbf{I}_d$ is the identity matrix of dimension $d$.*

**Lemma C.2** (Incoherence of Fourier Characteristics). *For $E \geq 1$, suppose $\mathbf{a}_1, \ldots, \mathbf{a}_E \overset{\text{iid}}{\sim} \mathcal{U}(\{-1, +1\}^{N\log_2(\mathcal{A})})$. Then,*

$$\mathbb{P}\left(\left\|\frac{\mathbf{X}^\top \mathbf{X}}{E} - I_{\mathcal{A}^N}\right\|_\infty \leq \sqrt{\frac{2\log\left(\frac{2\mathcal{A}^{2N}}{\delta}\right)}{E}}\right) \geq 1 - \delta,$$

where $\|\mathbf{A}\|_\infty = \max_{i,j} |\mathbf{A}_{i,j}|$ denotes the maximum coordinates of a matrix. Thus, if $E \geq 4096 \mathcal{A}^{2s} \left[ \log\left(\frac{2}{\delta}\right) + 2N \log\left(\mathcal{A}\right) \right]$, $\mathbf{X}$ is $\mathcal{A}^s$-incoherent with probability at least $1 - \delta$.

*Proof.* Recall that, for any $\mathbf{a} \in [\mathcal{A}]^N$, $\boldsymbol{\chi}(\mathbf{a}) := (\chi_{S_1}(\mathbf{a}), \dots, \chi_{S_{\mathcal{A}^N}}(\mathbf{a}))$, where $S_1, \dots, S_{\mathcal{A}^N}$ is some fixed enumeration of subsets $S \subset [N \log_2(\mathcal{A})]$. Thus, each entry of $(\mathbf{X}^\top \mathbf{X})/E$ can be viewed as being indexed by subsets $S, S' \subset [N \log_2(\mathcal{A})]$.

To establish (C.2), we first examine diagonal elements of $(\mathbf{X}^\top \mathbf{X})/E$. For $S \subset [N \log_2(\mathcal{A})]$, we have

$$\left(\frac{\mathbf{X}^\top \mathbf{X}}{E}\right)_{S,S} = \frac{1}{E}\left(\sum_{t=1}^E \boldsymbol{\chi}(\mathbf{a}_t)(\boldsymbol{\chi}(\mathbf{a}_t))^\top\right)_{S,S} = \frac{1}{E}\sum_{t=1}^E \chi_S(\mathbf{a}_t)\chi_S(\mathbf{a}_t) = 1, \qquad (3)$$

where the last equality follows from the fact that $(\chi_S(\mathbf{a}_t))^2 = 1$.

Next, we consider off-diagonal elements, and bound their magnitude. Before doing so, we require the following. For subsets $S, S' \subset [N \log_2(\mathcal{A})]$, let $S \Delta S'$ denote the symmetric difference of two subsets. For any two subsets $S, S' \subset [N \log_2(\mathcal{A})]$, the product of their Fourier characteristics is,

$$\chi_S(\mathbf{a}_t)\chi_{S'}(\mathbf{a}_t) = \left(\prod_{i \in S} \mathbf{v}(\mathbf{a}_t)_i\right)\left(\prod_{i' \in S'} \mathbf{v}(\mathbf{a}_t)_{i'}\right) = \prod_{i \in S \Delta S'} \mathbf{v}(\mathbf{a}_t)_i.$$

Using this, for any distinct subsets $S, S'$, we have

$$\left(\frac{\mathbf{X}^\top \mathbf{X}}{E}\right)_{S,S'} = \frac{1}{E}\sum_{t=1}^E \chi_S(\mathbf{a}_t)\chi_{S'}(\mathbf{a}_t) = \frac{1}{E}\sum_{t=1}^E \prod_{i \in S \Delta S'} \mathbf{v}(\mathbf{a}_t)_i.$$

Since $S \neq S'$, and $\mathbf{a}_t \sim \mathcal{U}(\{-1, 1\}^{N \log_2(\mathcal{A})})$, the set of random variables $\{\mathbf{v}(\mathbf{a}_t)_i : i \in S \Delta S'\}$ are independent Rademacher random variables. Applying Hoeffding's inequality for $\epsilon > 0$ gives us

$$\mathbb{P}\left(\left(\frac{\mathbf{X}^\top \mathbf{X}}{E}\right)_{S,S'} \geq \epsilon\right) \leq 2\exp\left(-\frac{E\epsilon^2}{2}\right).$$

Applying the inequality above and taking a union bound over all $\mathcal{A}^{2N}$ elements of $(\mathbf{X}^\top \mathbf{X})/E$

$$\mathbb{P}\left(\max_{\substack{S,S' \subset [N \log_2(\mathcal{A})] \\ S \neq S'}} \left(\frac{\mathbf{X}^\top \mathbf{X}}{E}\right)_{S,S'} \geq \epsilon\right) \leq 2\mathcal{A}^{2N}\exp\left(-\frac{E\epsilon^2}{2}\right).$$

Choosing $\epsilon = \sqrt{2E^{-1}\log\left(\frac{2\mathcal{A}^{2N}}{\delta}\right)}$ yields,

$$\mathbb{P}\left(\max_{\substack{S,S' \subset [N \log_2(\mathcal{A})] \\ S \neq S'}} \left(\frac{\mathbf{X}^\top \mathbf{X}}{E}\right)_{S,S'} \geq \sqrt{\frac{2\log\left(\frac{2\mathcal{A}^{2N}}{\delta}\right)}{E}}\right) \leq \delta. \qquad (4)$$

To complete the proof, observe that (3) implies that

$$\left\|\frac{\mathbf{X}^\top \mathbf{X}}{E} - I_{\mathcal{A}^N}\right\|_\infty = \max_{\substack{S,S' \subset [N \log_2(\mathcal{A})] \\ S \neq S'}} \left(\frac{\mathbf{X}^\top \mathbf{X}}{E}\right)_{S,S'}.$$

Substituting this observation into (4) above completes the proof. $\qquad \square$

In addition to the above lemma, we leverage the following Lasso convergence result. We state a version that can be found in the book on high-dimensional probability due to Rigollet and Hütter [2023].

**Lemma C.3** (Theorem 2.18 in [Rigollet and Hütter, 2023] ). *Suppose that $\mathbf{Y} = \mathbf{X}\theta^* + \epsilon$, where $\mathbf{X} \in \mathbb{R}^{E \times d}$, $\theta^* \in \mathbb{R}^d$ is s-sparse, and $\epsilon$ has independent 1-sub-Gaussian coordinates. Further, suppose $\frac{\mathbf{X}^\top \mathbf{X}}{T}$ is s-incoherent. Then, for any $\delta \in (0,1)$ and for $\lambda = 4\sqrt{E^{-1}\log(2d)} + 4\sqrt{E^{-1}\log(\delta^{-1})}$, we have, with probability at least $1 - \delta$*

$$\|\theta^* - \widehat{\theta}\|_2 \le C\sqrt{sE^{-1}\log\left(\frac{2d}{\delta}\right)}$$

*where $\widehat{\theta}$ denotes the solution to the Lasso and $C > 0$ is some absolute constant.*

Using standard arguments (see the proof of Theorem 2.18 in Rigollet and Hütter [2023] or the statement of Theorem 7.3 in Wainwright [2019]), it can be further deduced that $\|\theta^* - \widehat{\theta}\|_1 \le 4\sqrt{s}\|\theta^* - \widehat{\theta}\|_2$, so we actually have that, for any $\delta \in (0,1)$, with probability at least $1 - \delta$,

$$\|\theta^* - \widehat{\theta}\|_1 \le Cs\sqrt{E^{-1}\log\left(\frac{2d}{\delta}\right)},$$

where $C > 0$ is again some absolute constant.

## C.2 Proof of Theorem 5.1

*Proof.* Define $\theta = N^{-1}\sum_{n=1}^{N}\theta_n$. Recall the notation $\widehat{\theta} = N^{-1}\sum_{n=1}^{N}\widehat{\theta}_n$, and $\widehat{\mathbf{a}} = \arg\max_{\mathbf{a} \in [\mathcal{A}]^N}\langle\widehat{\theta}, \chi(\mathbf{a})\rangle$. For any round $t \in \{E+1, \dots, T\}$, we greedily play the action $\widehat{\mathbf{a}}$. The average reward $\bar{r}(\widehat{\mathbf{a}})$ can be bounded using the definition of $\widehat{\mathbf{a}}$ and Holder's inequality as follows,

$$\begin{aligned}
\bar{r}(\mathbf{a}^*) - \bar{r}(\widehat{\mathbf{a}}) &= \langle\theta, \chi(\mathbf{a}^*) - \chi(\widehat{\mathbf{a}})\rangle \\
&= \langle\theta - \widehat{\theta}, \chi(\mathbf{a}^*) - \chi(\widehat{\mathbf{a}})\rangle + \underbrace{\langle\widehat{\theta}, \chi(\mathbf{a}^*) - \chi(\widehat{\mathbf{a}})\rangle}_{\le 0} \\
&\le \langle\theta - \widehat{\theta}, \chi(\mathbf{a}^*) - \chi(\widehat{\mathbf{a}})\rangle \\
&\le \|\theta - \widehat{\theta}\|_1\|\chi(\mathbf{a}^*) - \chi(\widehat{\mathbf{a}})\|_\infty.
\end{aligned}$$

Next, substituting the definition of $\theta, \widehat{\theta}$, $\|\chi(\mathbf{a})\|_\infty = 1$, and using the triangle inequality into the equation above gives us,

$$\begin{aligned}
\bar{r}(\mathbf{a}^*) - \bar{r}(\widehat{\mathbf{a}}) &\le 2\|\theta - \widehat{\theta}\|_1 \le 2\left\|\frac{1}{N}\sum_{n=1}^{N}\left(\theta_n - \widehat{\theta}_n\right)\right\|_1 \\
&\le \frac{2}{N}\sum_{n=1}^{N}\left\|\theta_n - \widehat{\theta}_n\right\|_1.
\end{aligned} \tag{5}$$

Let us define the "good" events by

$$G_1 := \{\mathbf{X} \text{ is } \mathcal{A}^s\text{-incoherent}\} \quad \text{and} \quad G_2 := \left\{\forall n \in [N], \|\widehat{\theta}_n - \theta_n\|_1 \le C\mathcal{A}^s\sqrt{E^{-1}\log\left(\frac{4N\mathcal{A}^N}{\delta}\right)}\right\}$$

where $C > 0$ is the constant following the discussion of Lemma C.3. Let us define the global "good" event by $G := G_1 \cap G_2$. We show $\mathbb{P}(G) \ge 1 - \delta$.

First, there is a universal constant $C' > 0$ such that $E \ge 4096\mathcal{A}^{2s}[\log(4/\delta) + 2N\log(\mathcal{A})]$ when $T \ge C'\mathcal{A}^{2s}[\log(N/\delta) + N\log(\mathcal{A})]$. Thus, by Lemma C.2, we know the matrix $\mathbf{X}$ with $\chi^{\mathbf{a}_1}, \dots, \chi^{\mathbf{a}_E}$ as its rows is $\mathcal{A}^s$-incoherent least $1 - \frac{\delta}{2}$, i.e. $\mathbb{P}(G_1) \ge 1 - \frac{\delta}{2}$.

Next, conditioning on $G_1$ and applying Lemma C.3 alongside a union bound over the $N$ units yields

$$\|\widehat{\theta}_n - \theta_n\|_1 \le C\mathcal{A}^s\sqrt{E^{-1}\log\left(\frac{4N\mathcal{A}^N}{\delta}\right)},$$

for all $n \in [N]$ with probability at least $1 - \frac{\delta}{2}$, i.e. $\mathbb{P}(G_2 \mid G_1) \geq 1 - \frac{\delta}{2}$. Thus, in total, we have $\mathbb{P}(G) = \mathbb{P}(G_1)\mathbb{P}(G_2 \mid G_1) \geq (1 - \delta/2)^2 \geq 1 - \delta$. We assume we are operating on the good event $G$ going forward.

Plugging the per-unit $\ell_1$ norms into (5), the cumulative regret can be upper bounded with probability $1 - \delta$ as follows:

$$
\begin{aligned}
\mathrm{Reg}_T &= \sum_{t=1}^{T} \left( \bar{r}(\mathbf{a}^*) - \bar{r}(\hat{\mathbf{a}}) \right) \\
&= \sum_{t=1}^{E} \left( \bar{r}(\mathbf{a}^*) - \bar{r}(\hat{\mathbf{a}}) \right) + \sum_{t=E+1}^{T} \left( \bar{r}(\mathbf{a}^*) - \bar{r}(\hat{\mathbf{a}}) \right) \\
&\leq E + \frac{2T}{N} \left( \sum_{n=1}^{N} \left\| \boldsymbol{\theta}_n - \widehat{\boldsymbol{\theta}}_n \right\|_1 \right) \\
&\leq E + 2CT \cdot \mathcal{A}^s \sqrt{E^{-1} \log \left( \frac{4N\mathcal{A}^N}{\delta} \right)}
\end{aligned}
\tag{6}
$$

Clearly, we should select $E$ to roughly balance terms (up to multiplicative constants). In particular, using the choice of $E$ as

$$
E := (T\mathcal{A}^s)^{2/3} \left[ \log \left( \frac{N}{\delta} \right) + N \log(\mathcal{A}) \right]^{1/3}
$$

and substituting $E$ into (6) gives us

$$
\mathrm{Reg}_T \leq O \left( (N \log(\mathcal{A}N/\delta))^{1/3} (T\mathcal{A}^s)^{2/3} \right)
$$

with probability at least $1 - \delta$, precisely the claimed result. $\qquad \square$

