# OpenReview forum: "Mutli-Armed Bandits with Network Interference"
_NeurIPS.cc/2024/Conference — NeurIPS 2024 poster_

### Official Review · Reviewer_JzLc · 2024-06-15

**Soundness:** 4
**Presentation:** 3
**Contribution:** 3
**Rating:** 7
**Confidence:** 4

**Summary:**

The paper explores regret minimization in multi-armed bandits subject to interference. At each time, the learner assigns one of A treatments to each of the N arms. The observed per-arm response is subject to interference – ie, it depends on the assignments of all the other arms. This work studies the setting of s-sparse interference, where only the assignment of s "neighbors" of the arm affect its outcome. The paper's main theoretical contribution is to use discrete Fourier analysis to rewrite this problem as a linear bandit, then solve that bandit using tools from the linear bandits literature. An explore-then-commit algorithm is proposed that achieves a high-probability regret bound of (T A^s)^(⅔), with only logarithmic dependence on N.

**Strengths:**

This paper provides a novel formulation of bandits-with-interference as a linear bandits problem using discrete Fourier analysis. The problem is well-motivated by e-commerce applications, and the methods developed are to my knowledge novel. The paper is well-written, and provides thorough exposition of the results.

**Weaknesses:**

It took me a long time to realize that the effects were being estimated on a per-subset basis – ie, that chi^{a_i}(B(n)) is an indicator vector for the subset a_i intersected with the neighborhood of n. Originally I thought that chi^{a_i}(B(n)) =1 for all subsets of a_i, so I was very confused why we didn't have to worry about dependence among the entries of chi. If the authors can think of any way to clarify the exposition, I would greatly appreciate it. This is probably obvious to someone who is more familiar with Boolean Fourier analysis, but your readers may not be.

This paper is very well-written. One thing that was missing from the exposition was an explanation of what the Fourier analysis buys us in this setting. What did the Fourier approach allow you to do in this problem that was not available using previous methods (for example, methods people have applied to combinatorial bandits)? From reading your appendices, I think the answer might be that the orthogonality of the basis functions makes it easy to bound the precision of theta. If you could clarify this it would help your method be adopted by a wider audience.

**Questions:**

I'd like to make sure I understand the notation in this paper. Could the authors please provide, for a small toy example, numerical examples of the following vectors from Algorithm 1? $\bf{a_i}$ in Line 2, $\bf{\chi}(\bf{a_t})$ in Line 3, $\bf{\chi^{a_i}}(\mathcal{B}_n)$ in Line 5.

This paper makes a novel contribution of translating the "MAB-with-interference" problem to a linear bandits problem using Fourier analysis. I don't understand why you can't then use existing solutions from the linear bandits literature to solve your problem, and take regret rates from there, instead of rederiving the rates for your own algorithm. Could you please clarify why you cannot apply the linear bandits results mentioned in the introduction to your problem?

**Limitations:**

The model in the paper is extremely general: a unit's outcomes are dependent on the assignments of all of its neighbors, with no lower-order structure. As a result, the performance of the algorithm scales exponentially in the degree s. In reality I would expect the outcome to depend on lower-order interactions than s. I wonder if the authors have considered what would happen if they allowed the degree of the graph to exceed the order of interactions in the potential outcomes function, as in [1]. In this case a single assignment would give you information on multiple lower-order interaction coefficients. I imagine this would improve the performance of your algorithm significantly in the case where d << s, but the problem will get harder to analyze because the X vectors in your algorithm will now have correlations among the A^s elements of a single row. This will make bounding the singular values of X more difficult.

If you have space, I would love to see some discussion of how applicable your Fourier Analysis techniques are to these types of stronger modeling assumptions on the potential outcomes.

[1] Cortez, Mayleen, Matthew Eichhorn, and Christina Yu. "Staggered rollout designs enable causal inference under interference without network knowledge." Advances in Neural Information Processing Systems 35 (2022): 7437-7449.

---

> ### Author Rebuttal · Authors · 2024-08-06
>
> **Weaknesses:**
>
> _It took me a long time to realize that the effects were being estimated on a per-subset basis – ie, that chi^{a_i}(B(n)) is an indicator vector for the subset a_i intersected with the neighborhood of n. Originally I thought that chi^{a_i}(B(n)) =1 for all subsets of a_i, so I was very confused why we didn't have to worry about dependence among the entries of chi. If the authors can think of any way to clarify the exposition, I would greatly appreciate it. This is probably obvious to someone who is more familiar with Boolean Fourier analysis, but your readers may not be._
>
> Thank you for helping us improve the exposition of our paper! We provide a simple example of (a) binary action embeddings and (b) the corresponding Fourier embeddings below. We will also revise the paper to reflect this example.
>
> $\textbf{Binary action embedding}$. Consider $N$ units with 2 arms ($\mathcal{A} =2$), and a network graph $\mathcal{G}$ such that each unit $n$ is connected to one other unit, i.e., $\mathcal{G}$ has maximum degree 2. An action $\mathbf{a} = (a_{1}, \ldots a_{N}) \in \\{0,1\\}^N$, will induce the binary embedding $\mathbf{v}(\mathbf{a}) =  (v(a_1), \ldots v(a_N)) \in \\{-1,1\\}^N$ where $v(a_i) = 1$ if $a_i$ is 1 and -1 if $a_i = 0$. For example, if $\mathbf{a} = (0,1,1 \ldots 1)$, then $\mathbf{v}(\mathbf{a}) = (-1,1,\ldots,1)$.
>
> $\textbf{Fourier embedding}$. For any subset of units $ S \subset [N]$, the Fourier character $\chi_{S}(\mathbf{v}(\mathbf{a})) = \prod_{i \in S} v(a_i)$. For instance, if the action $\mathbf{a} = (0,1,1 \ldots 1)$ as above, and the subset $S = \\{1,2\\}$, the Fourier character  $\chi_{\\{1,2\\}}(\mathbf{v}(\mathbf{a}))  =  v(a_1) \times v(a_2) = -1$. The vector of Fourier characteristics $\boldsymbol{\chi}(\mathbf{a}) = (\chi_{S}(\mathbf{v}(\mathbf{a})) : S \subset N) \in \\{-1,1\\}^{2^N}$  is the concatenation of all Fourier characters for all subsets.
>
> $\textbf{Fourier coefficient}$. Since each unit $n$ is only connected to one other unit $m$, unit n's neighborhood $\mathcal{N}(n) = \\{n,m\\}$ for a unit $m \neq n$. For unit $n$, the blocks $\mathcal{B}(n)$ are the indices of $\mathbf{v}(\mathbf{a})$ corresponding to treatments of units in $\mathcal{N}(n)$. For our network graph $\mathcal{G}$, $\mathcal{B}(n) = \\{v(a_n), v(a_m)\\}$. The subsets $S \in \mathcal{B}(n) = \\{\phi, \\{n\\}, \\{m\\}, \\{n,m\\} \\}$ correspond to the non-zero coefficients of unit n's Fourier coefficient $\boldsymbol{\theta}_{n} \in \mathbb{R}^{2^{N}}$.
>
>
> _This paper is very well-written. One thing that was missing from the exposition was an explanation of what the Fourier analysis buys us in this setting. What did the Fourier approach allow you to do in this problem that was not available using previous methods (for example, methods people have applied to combinatorial bandits)? From reading your appendices, I think the answer might be that the orthogonality of the basis functions makes it easy to bound the precision of theta. If you could clarify this it would help your method be adopted by a wider audience._
>
> The Fourier basis provides a natural sparse linear representation of the network interference which is not easily done in other bases. To see this, we continue the example from the point above.  Since $\boldsymbol{\theta}_{n}$ is 4-sparse, the reward $r_n (\mathbf{a}) = \langle \boldsymbol{\theta}_n, \boldsymbol{\chi} (\mathbf{a}) \rangle$ for unit $n$ can be represented using 4 Fourier characters: $\\{\chi\_{\phi}, \chi\_{n}, \chi\_{m}, \chi\_{\\{n,m\\}} \\}$.  This representation captures sparsity unlike a 'one-hot' representation where the reward for unit $n$ can be represented as $r_n (\mathbf{a}) = \sum\_{\mathbf{a}' \in \\{0,1\\}^N}r_n(\mathbf{a}') \mathbf{1}[\mathbf{a} = \mathbf{a}']$. This one-hot basis also linearly expresses the reward but requires $2^N$ indicator basis vectors to do so.  It is precisely this sparse linear representation induced by the Fourier basis that allows our bounds to scale with $2^s$ rather than $2^N$.
>
> **Questions:**
>
> _This paper makes a novel contribution of translating the "MAB-with-interference" problem to a linear bandits problem using Fourier analysis. I don't understand why you can't then use existing solutions from the linear bandits literature to solve your problem, and take regret rates from there, instead of rederiving the rates for your own algorithm. Could you please clarify why you cannot apply the linear bandits results mentioned in the introduction to your problem?_
>
> We note that once we have mapped the network bandit problem into the corresponding Fourier basis, we still do not directly have a standard linear bandit problem. In particular, we are consider $N$ separate, simultaneous linear bandit problems (one for each unit), each with $\mathcal{A}^s$ possible actions. An additional aspect of our analysis is the aggregation of information over the various units to produce an estimate for the global average reward function.

---

> > ### Comment · Reviewer_JzLc · 2024-08-09
> >
> > Thank you for the detailed response.

---

### Official Review · Reviewer_Ry93 · 2024-06-30

**Soundness:** 3
**Presentation:** 3
**Contribution:** 2
**Rating:** 5
**Confidence:** 5

**Summary:**

The paper addresses the challenge of online experimentation in the presence of network interference, which is common in applications like e-commerce and clinical trials. The authors propose a multi-armed bandit (MAB) problem where a learner assigns actions (e.g., discounts) to
units (e.g., goods) over fixed and known rounds to minimize regret while considering the interference across units. However, naively applying canonical MAB methods (e.g., the upper-confidence-bound algorithm) will lead to regret with the exponential dependence on the number of units.

The main challenge of the paper is the exponentially large action space (each unit has |A| actions and thus |A|^n in total per round). The sparsity structure of the interference (each unit can only be affected by at most s neighboring units) implies that the effective action space for each unit can be a lot smaller. The paper designs an encoding scheme and an algorithm based on this observation. They show that the optimal regret in both $T$ and $|A|$ can be obtained.

**Strengths:**

I think the paper proposes an interesting problem to study. The problem is practical, well-motivated and mathematically elegant.The sparsity structure is also common and reasonable to assume. I enjoy reading the paper and learning about the problem.

The transformation to the linear space of functionals is interesting. It leads to the use of statistical bounds in high-dimensional statistics to be applied, which gives tight regret bounds.

I also like the use of LASSO to identify the interference strucuture. I believe this method may be of independent interest.

The algorithm and the theoretical results are presented clearly. The proof is written carefully and easy to read.

**Weaknesses:**

Although the formulation is interesting, the algorithm design and analyses seem quite standard. After transforming to the functional space, it becomes the MAB problem with effectively $|A|^s$ arms. The ETC algorithm and the analysis of the estimation error as well as the regret have been studied in the literature. I wonder if the formulation alone is sufficient as the contribution of the paper.

**Questions:**

The paper takes an agnostic approach to the network structure. This is the strength of the algorithm. However, I wonder if there is room for the network structure. For example, it doesn't seem possible for the algorithm to incorporate some network information such as clusters and stars. Is it possible to adapt the algorithm to certain graphs?

ETC is known to be not rate-optimal. I wonder why the authors don't use LinUCB algorithmsin linear bandit at the first place. The problem can be viewed as linear bandit with a very large space, correct?

In Step 10 of Algorithm 1, how can the optimization problem be solved in practice given the large space?

**Limitations:**

See **Questions**.

---

> ### Author Rebuttal · Authors · 2024-08-06
>
> **Weaknesses:**
>
> _Although the formulation is interesting, the algorithm design and analyses seem quite standard. After transforming to the functional space, it becomes the MAB problem with effectively $A^s$ arms. The ETC algorithm and the analysis of the estimation error as well as the regret have been studied in the literature. I wonder if the formulation alone is sufficient as the contribution of the paper._
>
> We believe there are several main contributions to our paper outside of the formulation, which we enumerate below.
> - First, as mentioned in the reviewer’s comment, the Fourier embedding of the unit-specific rewards is non-obvious and hasn’t been considered before in related literature.
> - Second, once we have performed the transformation using the discrete Fourier transform, the problem is not just a MAB instance with $A^s$ arms, but rather $N$ simultaneous multi-armed bandit instances, each having $A^s$ arms. Thus, an additional step we must take in our argument (and a step that isn’t considered in more classical approaches to ETC) is that we must aggregate our confidence sets for unit-specific reward functions into a corresponding confidence set for the global reward.
> - Third, we also provide a sequential elimination style algorithm which can obtain improved dependence on the time horizon $T$ in the regret bound. In particular, this style of algorithm gets the same dependence on $T$ as a UCB algorithm would.
>
> **Questions:**
>
> _The paper takes an agnostic approach to the network structure. This is the strength of the algorithm. However, I wonder if there is room for the network structure. For example, it doesn't seem possible for the algorithm to incorporate some network information such as clusters and stars. Is it possible to adapt the algorithm to certain graphs?_
>
> In our paper, we consider a minimal assumption setting in which all we know is a bound on the size of the neighborhood of each unit. We opt not to make additional structural assumptions on the network (e.g. clusters, stars, etc.) for the sake of simplicity and generality. We believe that better regret rates may be able to be achieved by making structural assumptions (for instance, in this case we may be able to regress against low-degree polynomials in the Fourier space), but this is outside of the scope of the paper.
>
> _ETC is known to be not rate-optimal. I wonder why the authors don't use LinUCB algorithms in linear bandit at the first place. The problem can be viewed as linear bandit with a very large space, correct?_
>
> In the setting of unknown network structure (Section 5 of our paper), the given dependence on the time horizon $T$ is optimal (i.e. no algorithm can generally achieve dependence better than $O(T^{2/3})$). See [1] for details, as the authors construct a lower bound. In the setting of known network interference (Section 4 of our paper), we actually provide a sequential-elimination algorithm (Algorithm 3 in Appendix D) that obtains optimal regret dependence on the time horizon T (a rate of $O(T^{1/2})$). While we could additionally consider a UCB-style algorithm, this would likely be more difficult to analyze, and wouldn’t improve the regret rate, at least with respect to the time horizon $T$ (sequential elimination and LinUCB achieve the same rate). We re-emphasize that, even after mapping to the frequency space, we still are not in a vanilla linear bandit problem — we are actually in a setting where there are $N$ simultaneous, related bandit instances.
>
> _In Step 10 of Algorithm 1, how can the optimization problem be solved in practice given the large space?_
>
> Computing the optimal action is equivalent to just finding the maximum in a large list (a trivial optimization problem for modern computers). We note that at this point in the algorithm there are no constraints on the selected action.
>
> [1] Botao Hao, Tor Lattimore, and Mengdi Wang. High-dimensional sparse linear bandits. Advances in Neural Information Processing Systems, 33:10753–10763, 2020.

---

> > ### Comment · Reviewer_Ry93 · 2024-08-09
> >
> > Thanks for the detailed response. I don't have further comments and remain positive of the paper.

---

> > > ### Author Response · Authors · 2024-08-12
> > >
> > > Thanks for the response. We believe we have addressed the primary concerns mentioned in the review. Is there anything else we can clarify to improve the score?

---

> > > > ### Comment · Reviewer_Ry93 · 2024-08-12
> > > >
> > > > I think I will keep my score at this point. If you believe one of the two points can be addressed, I'd be happy to raise the score. 1. The network structure and its impact on the regret. It doesn't have to be comprehenssive, but I believe it will strengthen the network side of the study. 2. For the computation of the optimal solution, although I understand that it just requires enumerating the action space $|A|^s$, I do have concerns about the exponential size of the action space itself. For a reasonablely sized problem, this would be quite impractical I think? What size of the problem can the solved by this method?

---

### Official Review · Reviewer_QVLr · 2024-07-10

**Soundness:** 4
**Presentation:** 3
**Contribution:** 3
**Rating:** 8
**Confidence:** 4

**Summary:**

The paper investigates a multi-armed bandit problem on a set of units that affect the rewards of each other. A trivial solution will have an exponential regret in the number of units, so using sparsity assumptions on the affecting neighborhood, the authors present an algorithm with regret that is only exponential in the sparsity coefficient. The authors present algorithms both for the case that the affecting neighborhood is known to the agent and the case that it is unknown.

**Strengths:**

* The paper presents and solves a useful problem in practice
* The theory is sound and the methods are original for network MAB, specifically learning the orthonormal coefficients is clever

**Weaknesses:**

On the one hand, Algorithm 1 performs well only when $N$ is very large (otherwise its better to use Algorithm 3), but it also has a running time of $\Omega(N)$, so it seems bad either way. Not sure why the focus is not on Algorithm 3 instead.

**Questions:**

* it seems you did not use the correct format for the submission (as their are no line numbers), make sure you correct this
* in the definition of "Linear Fourier expansion", should be $S \subset$ instead of $S \in$, right?
* In your algorithms, can you clarify the running time of finding the minimizing coefficient vector?
* In Theorem 4.1, should be $\mathcal{A}$ instead of $A$?

**Limitations:**

The authors properly address the limitations of their work.

---

> ### Author Rebuttal · Authors · 2024-08-06
>
> **Weaknesses:**
>
> _On the one hand, Algorithm 1 performs well only when $N$ is very large (otherwise its better to use Algorithm 3), but it also has a running time of $\Omega(N)$, so it seems bad either way. Not sure why the focus is not on Algorithm 3 instead._
>
> We chose to focus Section 4 on Algorithm 1 due to its simplicity and its similarity to the algorithm in the unknown interference structure case. We agree that Algorithm 3 likely offers a bound on regret in most parameter settings. For the final draft, we have added additional exposition about Algorithm 3 in Section 4 to better contrast the two approaches to regret minimization. In particular, we have added a theorem statement following the theorem statement for the “explore then commit” style algorithm.
>
>
> **Questions:**
>
> _Various typos and formatting issues_
>
> Thank you for pointing out the various typos and the issue with our paper's formatting. We have corrected these issues for the final draft of the paper.
>
> _In your algorithms, can you clarify the running time of finding the minimizing coefficient vector?_
>
> The runtime of finding the minimizing coefficient vector (i.e., $\hat{\mathbf{\theta}}_n$) is equivalent to solving an ordinary least squares or Lasso for $N$ units when the graph structure is known and unknown respectively. There exists efficient gradient-based algorithms for solving these linear programs. We note that our algorithms are designed such that each of these linear programs can be solved in parallel for each unit independently which significantly reduces runtime.

---

> > ### Comment · Reviewer_QVLr · 2024-08-09
> >
> > Thank you for the response, I will keep my score positive.

---

### Official Review · Reviewer_zUP1 · 2024-07-10

**Soundness:** 3
**Presentation:** 3
**Contribution:** 3
**Rating:** 6
**Confidence:** 3

**Summary:**

This article studies the multi-armed bandit (MAB) problem under unit interference. This unit inference problem is often considered in offline settings. This article extends it to online settings with a linear regression solution based on discrete Fourier features. Two Explore-Then-Commit algorithms are proposed to minimize regret under known and unknown interference, respectively. Finally, the algorithms are tested on some numerical simulations.

**Strengths:**

1. The paper is relatively well-written and easy to follow.
2. It is interesting to study the interference problem in online settings
3. The linear regression solution based on Fourier features seems novel.

**Weaknesses:**

1. The paper lacks real examples to demonstrate the combination of online experimentation and interference. Perhaps the combination is as natural as the paper suggests. For example, In online experimentation, after every action, it may take some time for the interference to occur.  If we measure the outcome right after the action, it may provide no information about the effect of interference on the reward.
2. The paper should discuss Assumption 2 more. For example, the upper bound s needs to hold for all units n. Suppose s is very large or small. It is unclear how this affects the algorithms presented later.
3. It is unclear if we can apply the offline methods under interference to online settings. Maybe we could use these methods to estimate the reward function at every step and then take action by maximizing the reward function.

**Questions:**

N.A.

**Limitations:**

N.A.

---

> ### Author Rebuttal · Authors · 2024-08-06
>
> **Weaknesses:**
>
> _The paper lacks real examples to demonstrate the combination of online experimentation and interference. Perhaps the combination is as natural as the paper suggests. For example, In online experimentation, after every action, it may take some time for the interference to occur. If we measure the outcome right after the action, it may provide no information about the effect of interference on the reward._
>
> A concrete example is given by online bidding in advertisement; the “agent” here is a centralized platform that coordinates at each round bids coming from N advertisers. Advertisers submit bids and compete in an auction: winning advertisers get to display the ad. Treatments here correspond to different pricing schemes imposed by the platform. For example, two different treatments might correspond to different reserve prices imposed on advertisers (e.g., a higher or lower premium paid at ad-display time). The reward function measures the downstream conversions driven by ads. Here, it is natural to assume that the reward for one advertiser will only be impacted by the behavior (impacted by the treatment) of a subset of the total population of advertisers, e.g., direct competitors.
>
> _The paper should discuss Assumption 2 more. For example, the upper bound s needs to hold for all units n. Suppose s is very large or small. It is unclear how this affects the algorithms presented later._
>
> Our algorithms leverage knowledge of the sparsity (or at least an upper bound on the sparsity s) in determining the length of the exploration period. As noted in Remark 4.6 in [1], we can select the length of the exploration period to be _independent_ of the sparsity, but we will pay an additional cost in regret of $O(A^{s/3})$. We have added a comment noting this in the appropriate sections. If one is concerned about practical applications of our algorithms, cross-validation can be used to select the length of the exploration period, as outlined in our remark at the end of Section 4. We also note that such sparsity assumptions are common in both network causal inference literature and the high-dimensional statistics literature.
>
> _It is unclear if we can apply the offline methods under interference to online settings. Maybe we could use these methods to estimate the reward function at every step and then take action by maximizing the reward function._
>
> We note that the goals of problems considered in the offline setting are not the same as those considered in the online setting. In the offline setting, the goal of the learner is typically to (a) estimate some sort of treatment effect in the presence of network interference and (b) produce a confidence interval/set for the parameter estimate. The experiments designed to accomplish these tasks often involve uniform/random exploration over the entire time period, and thus would yield linear regret. In the online setting, we want to minimize regret. While we do estimate the underlying global reward functions, this is just a nuisance parameter in our setting. What we really care about is (a) quickly discovering a (nearly) action and (b) exploiting this action over many time steps. This difference is what necessitates different algorithms.
>
> [1] Botao Hao, Tor Lattimore, and Mengdi Wang. High-dimensional sparse linear bandits. Advances in
> Neural Information Processing Systems, 33:10753–10763, 2020.

---

### Official Review · Reviewer_RoeT · 2024-07-12

**Soundness:** 4
**Presentation:** 3
**Contribution:** 3
**Rating:** 7
**Confidence:** 3

**Summary:**

This paper introduces a Multi-Armed Bandits framework to address the challenge of online experimentation with network effects. Specifically, the authors consider a learner sequentially assigning one type of $\mathcal{A}$ actions to $N$ units over $T$ periods to minimize the regret. The reward from each unit depends not only on the action it received but also on the actions its neighbors received, i.e., there is interference across the underlying network of units. The contribution of this paper is as follows:

- Using Boolean encoding and Fourier series of Boolean functions, the authors re-express the reward function of each unit as a linear function of the Fourier basis. Then, they propose a simple 'explore-then-commit' style algorithm to address the challenge of the MAB problem with network interference.

- With known interference, i.e., the underlying neighbors of each unit are known by the learner, the authors show that their proposed 'explore-then-commit' type algorithm possesses a sublinear regret in $T$ and $N$.

- With unknown interference, the authors use LASSO to estimate the parameters on the Fourier basis, and establish a similar sublinear regret. The authors also argue the scaling of $T$ in their regret bound cannot be improved.

- Numerical simulations validate the effectiveness of the proposed algorithms and show they outperform the UCB baseline.

**Strengths:**

- The paper is very clear and well-written.
- The analysis is sound and well-discussed for all the limitations
- The discrete Fourier decomposition requires a deep understanding and keen observation of the problem.
- Although the algorithm follows a simple 'explore-then-commit' style, the analysis is non-trivial and possesses theoretical and technical difficulties.
- The regret is both sublinear in $N$ and $T$, and also $\mathcal{A}^s$ where $s$ is the degree of neighbors.

**Weaknesses:**

- I feel it's not common to see $\mathcal{A}$ to denote a number rather than the action set.
- The Boolean encoding of the actions is not well-discussed.
- In Fig 2(b), it seems that the regret stops accumulating when $T$ is larger enough. Is it true? Could you give a discuss why it happens?
- There are some typos of notations, for example, on page 4, (2) Simple orthonormal basis, the authors used $\mathcal{F}_{bool}$.

**Questions:**

The authors studied the interference over the underlying network of units, I wonder if it's possible to consider the impact over time, i.e., the reward may rely on previous actions that its neighbors received, for example, the impact decays exponentially over time.

**Limitations:**

The limitations are well addressed.

---

> ### Author Rebuttal · Authors · 2024-08-06
>
> **Weaknesses:**
>
> _The Boolean encoding of the actions is not well-discussed._
>
> Thank you for helping us improve the exposition of our paper! We provide a simple example of (a) binary action embeddings and (b) the corresponding Fourier embeddings below. Further, we describe how the Fourier representation naturally captures our sparse network interference assumption unlike other potential representations. This example can be found in the response to all reviewers above, but we discuss it here for convenience. We will also revise the paper to reflect this example.
>
> $\textbf{Binary action embedding}$. Consider $N$ units with 2 arms ($\mathcal{A} =2$), and a network graph $\mathcal{G}$ such that each unit $n$ is connected to one other unit, i.e., $\mathcal{G}$ has maximum degree 2. An action $\mathbf{a} = (a_{1}, \ldots a_{N}) \in \\{0,1\\}^N$, will induce the binary embedding $\mathbf{v}(\mathbf{a}) =  (v(a_1), \ldots v(a_N)) \in \\{-1,1\\}^N$ where $v(a_i) = 1$ if $a_i$ is 1 and -1 if $a_i = 0$. For example, if $\mathbf{a} = (0,1,1 \ldots 1)$, then $\mathbf{v}(\mathbf{a}) = (-1,1,\ldots,1)$.
>
> $\textbf{Fourier embedding}$. For any subset of units $ S \subset [N]$, the Fourier character $\chi_{S}(\mathbf{v}(\mathbf{a})) = \prod_{i \in S} v(a_i)$. For instance, if the action $\mathbf{a} = (0,1,1 \ldots 1)$ as above, and the subset $S = \\{1,2\\}$, the Fourier character  $\chi_{\\{1,2\\}}(\mathbf{v}(\mathbf{a}))  =  v(a_1) \times v(a_2) = -1$. The vector of Fourier characteristics $\boldsymbol{\chi}(\mathbf{a}) = (\chi_{S}(\mathbf{v}(\mathbf{a})) : S \subset N) \in \\{-1,1\\}^{2^N}$  is the concatenation of all Fourier characters for all subsets.
>
> $\textbf{Fourier coefficient}$. Since each unit $n$ is only connected to one other unit $m$, unit n's neighborhood $\mathcal{N}(n) = \\{n,m\\}$ for a unit $m \neq n$. For unit $n$, the blocks $\mathcal{B}(n)$ are the indices of $\mathbf{v}(\mathbf{a})$ corresponding to treatments of units in $\mathcal{N}(n)$. For our network graph $\mathcal{G}$, $\mathcal{B}(n) = \\{v(a_n), v(a_m)\\}$. The subsets $S \in \mathcal{B}(n) = \\{\phi, \\{n\\}, \\{m\\}, \\{n,m\\} \\}$ correspond to the non-zero coefficients of unit n's Fourier coefficient $\boldsymbol{\theta}_{n} \in \mathbb{R}^{2^{N}}$.
>
> $\textbf{Fourier basis captures sparsity}$.  Since $\boldsymbol{\theta}_{n}$ is 4-sparse, the reward $r_n (\mathbf{a}) = \langle \boldsymbol{\theta}_n, \boldsymbol{\chi} (\mathbf{a}) \rangle$ for unit $n$ can be represented using 4 Fourier characters: $\\{\chi\_{\phi}, \chi\_{n}, \chi\_{m}, \chi\_{\\{n,m\\}} \\}$.  This representation captures sparsity unlike a 'one-hot' representation where the reward for unit $n$ can be represented as $r_n (\mathbf{a}) = \sum\_{\mathbf{a}' \in \\{0,1\\}^N}r_n(\mathbf{a}') \mathbf{1}[\mathbf{a} = \mathbf{a}']$. This one-hot basis also linearly expresses the reward but requires $2^N$ indicator basis vectors to do so.
> It is precisely this sparse linear representation induced by the Fourier basis that allows our bounds to scale with $2^s$ rather than $2^N$.
>
>
> _In Fig 2(b), it seems that the regret stops accumulating when T is larger enough. Is it true? Could you give a discuss why it happens?_
>
> The regression algorithm (either OLS in the case of Algorithm 1 or the Lasso in the case of Algorithm 2) estimates the unknown global reward function with a high-degree of accuracy at the end of the exploration phase. In more detail, we have that (with high probability) $|\widehat{r}(a) - \bar{r}(a)| < \epsilon$ for all actions $a$, where $\epsilon$ is some small value denoting the width of the confidence interval. In the case that the suboptimality gap (i.e. the gap in reward between the best and second best action) is smaller than $\epsilon$, we are guaranteed to select the optimal action. That is, we incur zero regret during the exploitation phase.
>
>
> **Questions:**
>
> _The authors studied the interference over the underlying network of units, I wonder if it's possible to consider the impact over time, i.e., the reward may rely on previous actions that its neighbors received, for example, the impact decays exponentially over time._
>
> We believe investigating the effects of an entire history of treatments on the current period is both interesting and practically-relevant. Given that the current approach for estimating rewards depends heavily on the unknown, unit-specific reward functions being fixed over the rounds of interaction, additional machinery would almost surely need to be developed to handle this generalization. Here is perhaps one possible approach:
> -  If the dependence of present reward (say in round $t$) on historical treatments decays geometrically, it may be possible to “extend” the action space to tuples of actions played over the past $s$ rounds (i.e. tuple of the form $(a_t, a_{t - 1}, \dots, a_{t - s})$. We can then suffer some “truncation” error for ignoring the given treatments in rounds $t -s -1, t-s -2$, etc. This truncation error can be computed based on the geometric rate of decay, and $s$ can likely be chosen with respect to the time horizon $T$ to obtain small regret. Additionally, machinery from the reinforcement learning literature may be applicable to handling time-dependent rewards. We note that both aforementioned approaches fall outside of the scope of this paper, and thus we leave them for future work.

---

> > ### Comment · Reviewer_RoeT · 2024-08-12
> >
> > Thanks for the response. I will keep my positive score.

---

### Official Review · Reviewer_ctjH · 2024-07-18

**Soundness:** 3
**Presentation:** 4
**Contribution:** 3
**Rating:** 7
**Confidence:** 2

**Summary:**

The paper tackles a multi-armed bandit problem where there is interference that can be modeled by a network. The interference model assumes that a unit's treatment effect is affected by the treatments assigned to its neighbors in the network model. This interference model implies that in the worst case there are $\mathcal{A}^{N}$ possible combination of arms to be pulled each round which makes the regret minimization analysis difficult using existing techniques. To tackle this challenge, the paper studies the sparse network interference model and uses discrete Fourier analysis to show the unit-specific reward can learned using sparse linear regression-based algorithms. They provide a regret minimization algorithm for the setting where the interference model is explicitly known and for the setting where it is unknown. They conclude the paper numerical simulations to corroborate their theoretical findings.

**Strengths:**

The paper is well-written with ample discussion of the background, notation, set-up, theorems, and relation to existing work. As a result the paper is easy to understand and able to highlight its contributions.

Contribution-wise, the paper considers a relevant multi-armed bandit with interference setting and is able to consider a richer class of actions compared existing work. The assumption that the interference network is sparse seems reasonable and intuitive given the real-world settings discussed in the paper. The regret bounds presented seem intuitive and are able to effectively leverage the sparse network structure. The paper's contributions provide a good generalization of network interference that can be used in future work.

**Weaknesses:**

One potential weakness of the paper is that it provides limited proof sketches on results related leveraging the sparse network. Intuition on how the proof technique works may be useful for readers interested in utilizing similar assumptions for future research.

The paper also only briefly touches upon settings where the graph is partially observed. The prescribed solution in these settings is to use the fully observed algorithm if "all" the neighbors of the unit are observed and to use the unobserved algorithm if the neighbors are not observed. It still seems unclear what should happen if only "some" of the neighbors are observed. Perhaps additional detail can be added to clarify.

**Questions:**

1. Does the sparse network model effectively capture interference effects that decay depending on how related two units are? For example, unit $i$ is weakly related to units $1, \dots, K$ so that the arm assignment for one these units has a small effect, but aggregated over all $K$ units has a large effect.
2. If only a portion of neighbors is revealed for unit $i$, do you use Algorithm 1 or Algorithm 2?

**Limitations:**

Authors have adequately addressed limitations.

---

> ### Author Rebuttal · Authors · 2024-08-06
>
> **Weaknesses:**
>
> _One potential weakness of the paper is that it provides limited proof sketches on results related leveraging the sparse network._
>
> We agree that a proof sketch would help the reader better understand our algorithm and its convergence. If accepted, we will use the additional page to add brief proof sketches following the theorem statements.
>
> _The paper also only briefly touches upon settings where the graph is partially observed. The prescribed solution in these settings is to use the fully observed algorithm if "all" the neighbors of the unit are observed and to use the unobserved algorithm if the neighbors are not observed. It still seems unclear what should happen if only "some" of the neighbors are observed. Perhaps additional detail can be added to clarify._
>
> If only some of the neighbors are observed, or if the practitioner has any doubt about whether a given edge is present in the graph/network, it is likely best practice to use the Algorithm 2 (i.e. the algorithm for the fully-unobserved graph case). We note that both in theory and in our simulations, the performance of this algorithm is not much worse than that of Algorithm 1. Moreover, we emphasize that neither the fully or partially observed settings have been studied before this paper. While it may be possible to derive algorithms for particular forms of unobserved structure, it falls outside of the scope of our work. We have added clarification about this point in the paper.
>
> **Questions:**
>
> _Does the sparse network model effectively capture interference effects that decay depending on how related two units are?_
>
> Our paper considers a worst-case setting in which the reward/outcomes associated with each unit can depend arbitrarily on the treatments assigned to its neighbors. We believe that better regret rates are possible if additional structure is assumed, e.g. that the reward satisfies a “bounded difference” property and is not very sensitive to a change in treatment for any given individual unit. Deriving algorithms that can leverage additional structural assumptions is likely a non-trivial task, so we leave it for future work.
>
> _If only a portion of neighbors is revealed for unit i, do you use Algorithm 1 or Algorithm 2?_
>
> In this case, the learner should use Algorithm 2.

---

> > ### Comment · Reviewer_ctjH · 2024-08-12
> >
> > I appreciate the response and the clarifications. I will keep my positive score.

---

### Author Rebuttal · Authors · 2024-08-06

We would like to thank the reviewers for the time spent reviewing our work. We greatly appreciate the feedback and will use it to improve our work.

We want to clarify that we view our primary contributions to be the following.
- A framework for studying multi-armed bandits in the presence of network interference under minimal structural assumptions.
- An embedding our this framework into a setting of $N$ parallel linear MAB instances using aspects of discrete Fourier analysis.
- Simple explore-then-commit and sequential elimination style algorithms that can be used to obtain small regret.


In addition, several reviewers noted that our exposition on the aspects of Fourier analysis necessary for our results was a bit dense. We provide a simple example of (a) binary action embeddings and (b) the corresponding Fourier embeddings below. Further, we describe how the Fourier representation naturally captures our sparse network interference assumption unlike other potential representations. This example can be found in the reviews below, but we discuss it here for convenience. We will also revise the paper to reflect this example.

$\textbf{Binary action embedding}$. Consider $N$ units with 2 arms ($\mathcal{A} =2$), and a network graph $\mathcal{G}$ such that each unit $n$ is connected to one other unit, i.e., $\mathcal{G}$ has maximum degree 2. An action $\mathbf{a} = (a_{1}, \ldots a_{N}) \in \\{0,1\\}^N$, will induce the binary embedding $\mathbf{v}(\mathbf{a}) =  (v(a_1), \ldots v(a_N)) \in \\{-1,1\\}^N$ where $v(a_i) = 1$ if $a_i$ is 1 and -1 if $a_i = 0$. For example, if $\mathbf{a} = (0,1,1 \ldots 1)$, then $\mathbf{v}(\mathbf{a}) = (-1,1,\ldots,1)$.

$\textbf{Fourier embedding}$. For any subset of units $ S \subset [N]$, the Fourier character $\chi_{S}(\mathbf{v}(\mathbf{a})) = \prod_{i \in S} v(a_i)$. For instance, if the action $\mathbf{a} = (0,1,1 \ldots 1)$ as above, and the subset $S = \\{1,2\\}$, the Fourier character  $\chi_{\\{1,2\\}}(\mathbf{v}(\mathbf{a}))  =  v(a_1) \times v(a_2) = -1$. The vector of Fourier characteristics $\boldsymbol{\chi}(\mathbf{a}) = (\chi_{S}(\mathbf{v}(\mathbf{a})) : S \subset N) \in \\{-1,1\\}^{2^N}$  is the concatenation of all Fourier characters for all subsets.

$\textbf{Fourier coefficient}$. Since each unit $n$ is only connected to one other unit $m$, unit n's neighborhood $\mathcal{N}(n) = \\{n,m\\}$ for a unit $m \neq n$. For unit $n$, the blocks $\mathcal{B}(n)$ are the indices of $\mathbf{v}(\mathbf{a})$ corresponding to treatments of units in $\mathcal{N}(n)$. For our network graph $\mathcal{G}$, $\mathcal{B}(n) = \\{v(a_n), v(a_m)\\}$. The subsets $S \in \mathcal{B}(n) = \\{\phi, \\{n\\}, \\{m\\}, \\{n,m\\} \\}$ correspond to the non-zero coefficients of unit n's Fourier coefficient $\boldsymbol{\theta}_{n} \in \mathbb{R}^{2^{N}}$.

$\textbf{Fourier basis captures sparsity}$.  Since $\boldsymbol{\theta}_{n}$ is 4-sparse, the reward $r_n (\mathbf{a}) = \langle \boldsymbol{\theta}_n, \boldsymbol{\chi} (\mathbf{a}) \rangle$ for unit $n$ can be represented using 4 Fourier characters: $\\{\chi\_{\phi}, \chi\_{n}, \chi\_{m}, \chi\_{\\{n,m\\}} \\}$.  This representation captures sparsity unlike a 'one-hot' representation where the reward for unit $n$ can be represented as $r_n (\mathbf{a}) = \sum\_{\mathbf{a}' \in \\{0,1\\}^N}r_n(\mathbf{a}') \mathbf{1}[\mathbf{a} = \mathbf{a}']$. This one-hot basis also linearly expresses the reward but requires $2^N$ indicator basis vectors to do so. It is precisely this sparse linear representation induced by the Fourier basis that allows our bounds to scale with $2^s$ rather than $2^N$.

---

### Decision · Program_Chairs · 2024-09-25

**Decision:**

Accept (poster)

**Comment:**

I concur with the reviewers' positive view of the paper and think it will make a great addition to the nascent literature on this emerging interesting problem. I would like to point out to the authors one important misunderstanding regarding the difference between the adversarial and stochastic bandit problems when comparing to the existing literature, and I would like to ask them to fix this when preparing their camera ready version. The setting studied herein has an important difference to that that of Jia et al: in this paper the mean reward is both stationary and stochastic ($r_n(a_1,\dots,a_n)$ is a fixed function that does not depend on $t$ and $\epsilon_{nt}$ is a random variable) whereas in Jia et al the rewards are non-stationary and can be a potentially adversarial (subject to a decaying interference constraint) non-random sequence (written therein as $Y_{ut}$ over $u=1,\dots,N$; the only randomness involved in this problem and the regret analysis is from the user choice of actions). In this way, when $N=1$, the present paper reduces to the standard stochastic bandit (where eg UCB is used) while Jia et al reduces to the standard adversarial bandit (where eg Exp3 is used), hence the problems have quite a different formalism and inherent nature. Therefore, it is not quite correct to say "These results significantly improve upon
previous works on this topic, which impose stronger conditions on a known interference network,
and compare regret to a markedly weaker policy" (page 2). In particular, in the adversarial setup there can be little hope for the regret against the unit-adaptive policy and all the emerging non-trivial phenomena are exactly because one compares to the unit-constant policy. Given that this is an emerging problem it would be great for the authors to discuss and explain these connections/differences so the reader can fully understand the emerging landscape well. I trust this is do'able within the scope of preparing a camera ready as it only involves small edits and adding some more discussions.